# Not just words! Effects of a light-touch randomized encouragement intervention on students' exam grades, self-efficacy, motivation, and test anxiety

Tamás Keller  [1,2,3]*, Péter Szakál[4]

**1** Centre for Social Sciences: Research Center for Educational and Network Studies, Budapest, Huangary,
**2** Research Center for Economic and Regional Studies: Institute of Economics, Budapest, Huangary,
**3** TÁRKI Social Research Institute, Budapest, Huangary, **4** University of Szeged, Szeged, Huangary

* keller.tamas@tk.hu

Effects of a light-touch randomized encouragement
intervention on students' exam grades, self-
efficacy, motivation, and test anxiety. PLoS ONE
16(9): e0256960. https://doi.org/10.1371/journal.
pone.0256960

Murcia, SPAIN

**Data Availability Statement:** We archived data and
analytic scripts on the project's page on the Open
Science framework: https://osf.io/qkfe4/.

## Abstract

Motivated by the self-determination theory of psychology, we investigate how simple school
practices can forge students' engagement with the academic aspect of school life. We car-
ried out a large-scale preregistered randomized field experiment with a crossover design,
involving all the students of the University of Szeged in Hungary. Our intervention consisted
of an automated encouragement message that praised students' past achievements and
signaled trust in their success. The treated students received encouragement messages
before their exam via two channels: e-mail and SMS message. The control students did not
receive any encouragement. Our primary analysis compared the treated and control stu-
dents' end-of-semester exam grades, obtained from the university's registry. Our secondary
analysis explored the difference between the treated and control students' self-efficacy,
motivation, and test anxiety, obtained from an online survey before students' exams. We did
not find an average treatment effect on students' exam grades. However, in the subsample
of those who answered the endline survey, the treated students reported higher self-efficacy
than the control students. The treatment affected students' motivation before their first exam
—but not before their second—and did not affect students' test anxiety. Our results indicate
that automated encouragement messages sent shortly before exams do not boost students'
exam grades, but they do increase self-efficacy. These results contribute to understanding
the self-efficacy mechanism through which future encouragement campaigns might exert
their effect. We conclude that encouraging students and raising their self-efficacy might cre-
ate a school climate that better engages students with the academic aspect of school life.

## I. Introduction

Students' engagement with the academic aspect of school life is based on positive emotional
involvement in initiating and carrying out learning activities [1]. Engaged students develop
skills and abilities that help them to adjust to school: they maintain positive beliefs about their

**Funding:** This research was funded by a grant from the Hungarian National Research, Development and Innovation Office (NKFIH), Grant number: K-135766 to Tamás Keller. Tamás Keller acknowledges the support from the János Bolyai Research Scholarship of the Hungarian Academy of Sciences (BO/00569/21/9) and the New National Excellence Program (ÚNKP) of the Ministry of Human Capacities (Grant Number: ÚNKP-21-5-CORVINUS-132).

**Competing interests:** The authors have declared that no competing interests exist.

competence, are self-determined, and report a low level of anxiety [2]. Therefore, students' engagement affects school achievement [3] and is one of the major components in understanding dropout and promoting school completion [4].

The self-determination theory explicates the motivational foundation of students' engagements and posits that self-motivated and self-determined behavior hinges on fulfilling fundamental needs of autonomy, competence, and relatedness [5, 6]. The theory points out that contextual factors under schools' control can facilitate student self-determination and promote students' engagement with school expectations [3]. By contrast, if schools have deficient practices that lead to unsuccessful school outcomes, this decreases students' self-esteem and ensures problematic behaviors that further encourage unsuccessful school outcomes [7]. In short, specific school practices can foster engaging school climates. Supportive school practices are especially important in older age, when students have already accumulated some bad experiences that they need to overcome [3].

This paper investigates a particular school practice introduced on an experimental basis at a Hungarian university (the University of Szeged) to develop a student-friendly university climate. We investigated (1) whether a light-touch intervention—an automated encouragement message—can induce an exogenous change in students' ability-beliefs, and (2) how much the induced change translates to a gain in students' school performance measured by end-of-semester exam grades. We focus on three specific beliefs that express students' perceptions of their ability to some extent.

The first belief we focus on is self-efficacy: a persons' confidence in their own ability to complete a particular task [8]. Students' self-efficacy in regulating their learning and mastering their academic activities determines their aspirations and level of motivation [9]. It activates students' belief in their competence [10], fuels their expectancy of success [11], regulates the amount of effort students invest in a given task, and determines how long they persevere [12]. Therefore, self-efficacy directly influences students' learning outcomes [13]. Furthermore, research in educational psychology has shown that self-efficacy reduces emotional stress and might have a beneficial indirect effect on students' performance [14].

The second belief is the motivational belief in students' own readiness to perform a given behavior. This belief ultimately rests on trust in one's own ability. In his seminal work, Ajzen [15] describes a similar concept—behavioral intention—which hinges on the perceived control over the intended behavior. In Ajzen's theory, behavioral intention regulates how hard students try and how much effort they exert in performing a goal. Therefore, students who intend to succeed in an exam may, in fact, be more likely to achieve success, since the stronger the intention to engage in a behavior, the more likely its realization.

The last belief we focus on is test anxiety, which is a worrisome belief students hold about their own failure [16], fueled by negative beliefs about their own ability [17]. Test anxiety hinders individual learning and blocks students from presenting already acquired knowledge. Test anxiety, therefore, reduces academic performance [18], as worrying about failure prevents students from concentrating on the exam [19].

Our interest in evaluating the effects of a treatment targeting these beliefs is motivated by research in economics and educational psychology showing that beliefs related to academic success are malleable [20, 21]. Various interventions have successfully improved students' school performance by developing their mindfulness [22], social skills [23], social-emotional competencies [24], or self-concept [25].

Nevertheless, in educational practice, the implemented programs differ in intensity. For example, the 2-year long xl club program focused on improving students' confidence, self-esteem, and motivation [26]. The intensive development of these skills implemented in small groups brought about a rise in these skills in the program.

Alongside long and intensive interventions, simple behavioral procedures like the Good Behavior Game [27, 28] successfully spur students' self-regulation by introducing regular routines in the daily operation of education. Participants in the program scored higher test scores in reading and mathematics than students in the matched control group [29].

Nevertheless, light-touch encouragements can also induce a change in students' test results, particularly by targeting self-confidence and test anxiety. Behncke conducted a small randomized trial at the University of St. Gallen in Switzerland and revealed that students whose teacher read aloud a standard positive affirmation message before their exam scored higher in tests than those who had not received the positive affirmation [30]. Furthermore, Deloatch et al. [31] documented that highly test-anxious students who could read their Facebook friends' affirmation messages before an exam situation scored similarly to low test-anxious students.

Still, there are at least three concerns that prevent the overgeneralization of these positive results. First, prior meta-analyses show significant heterogeneity in the effect sizes; larger studies report a smaller effect size [23]. Programs introduced in education are particularly prone to a negative correlation between sample size and effect size [32]. Therefore, well-executed large-scale studies that employ an experimental design and impact students' achievement via their noncognitive skills often report limited or no findings [33, 34]. This suggests that small case studies are insufficient to determine a particular educational program's scientific validity and practical utility. Therefore, upcoming large-scale studies should corroborate the explorative results of small-scale experiments and produce conclusive evidence of the effectiveness of a given program.

Second, the efficacy of the developmental programs in education hinges on teachers' understanding of the program and their capacity to implement it [35]. These programs either require a change in teachers' daily school routines or endow teachers with new skills. Altering teachers' daily school routines can increase teachers' workload. Teachers may thus become less motivated to implement these programs, ultimately inhibiting the program's efficacy. Integrating developmental programs into teachers' training systems and thus endowing teachers with new skills slow down the interventions' return process [36]. Only a scant number of studies propose light-touch interventions that are ready to be integrated into educational practice without requiring teachers' motivation or experience.

Third, studies often fail to detect the particular belief or (non-cognitive) skill that could potentially induce the change in the targeted cognitive skills [37]. This shortcoming is especially problematic if the intervention does not directly influence students' cognitive skills. The lack of knowledge about the treatment mechanisms could lead to an underrating of the programs' general importance, making it more difficult for future research to improve the intervention [26].

This paper advances our understanding of each of these concerns. First, we have conducted a large-scale, well-powered, and preregistered randomized field experiment that involved all the students of the University of Szeged (N = 15,539) in Hungary. Thus, our study is not specific to a particular subpopulation of students but is well powered to detect small effect sizes and capable of exploring treatment heterogeneity.

Second, we have developed an easily scalable light-touch intervention that does not require teachers' attention. Students received an encouragement message before their exam—via e-mail and SMS text message—from the Head of the Directorate of Education at the university.

Third, we focus on particular mechanisms proposed by Behncke [30]: self-efficacy, motivation, and test anxiety. Identifying the treatment mechanisms promotes innovative and more effective future treatments [38].

Specifically, our intervention consisted of an automated message that the treated students received before their exam. The language of the message praised students' past achievements and signaled trust in their success. Thus, we targeted students' ability beliefs by empowering them. We randomized whether students received the treatment before their first or second exam. Therefore, we could observe each student when they received and did not receive the treatment, enabling us to compare students to themselves under different conditions.

We evaluated the treatment's effect on our primary outcome—end-of-semester exam grades, which we assessed from the university's register. Furthermore, we investigated the treatment effect in various secondary outcomes such as self-efficacy, motivation, and test anxiety. These measures were collected via an online survey that both treated and control students filled in before the exam, and thus data on the secondary outcomes are available for a subsample of the students.

Our results show that the encouragement message had no effect on students' average exam grades (primary outcome) in the whole sample. Initially more able students, however, did achieve higher grade scores if they were encouraged.

Out of our three secondary outcomes, we find a positive treatment effect in one outcome variable (self-efficacy). Specifically, treated students reported higher self-efficacy than control students. Concerning the two other secondary outcomes: in the case of students' motivation, the treatment effect is most evident in students' first exam but is attenuated in their second exam. The treatment did not translate into a significant decrease in students' test anxiety.

In sum, light-touch, automated encouragement messages, requiring minimal additional human effort from the message provider and sent shortly before exams, do not affect students' exam grades. Nevertheless, we have isolated a possible mechanism through which encouragement interventions might exert their effect. Specifically, we found that self-efficacy is sensitive to encouraging words, even if students only receive them on an occasional basis shortly before an academically challenging exam situation.

Future encouragement interventions should therefore use innovative and effective encouragement messages that target students' self-efficacy. We have two recommendations for future research. First, a personalized encouragement message sent by a sender to whom students have contact might be worth considering. Second, future interventions should consider encouraging students systematically and not just shortly before their exams.

We conclude that encouraging students has its own value even if it is not the appropriate tool to increase students' average exam grades. Receiving empowerment from the university contributes to feelings of importance and acknowledgment that are necessary factors in preventing university dropout [39]. Furthermore, prior research argues that many students leave university after their perception of their ability is affected by their recently awarded grades [40]. Providing positive feedback to students may thwart these processes and contribute to a school climate that engages students. Our results suggest that by increasing self-efficacy, the encouragement of students could contribute to a potential school practice that forges positive emotional involvement and engagement with the academic aspect of school life [3, 4].

## II. Design, data, and method

### II. 1. Preregistration

Our coding choices and statistical analysis closely follow our detailed pre-analysis plan, which we archived at the registry for randomized controlled held by the American Economic Association (https://doi.org/10.1257/rct.5155-1.1) before the end of the fieldwork and before receiving any kinds of endline data. Deviations from the preregistered pre-analysis plan is listed in the S10 Appendix.

We archived supplementary materials, data and analytic scripts on the project's page on the Open Science framework: https://osf.io/qkfe4/. The study was reviewed and approved by the IRB office at the Centre for Social Sciences, Budapest.

## II. 2. The field experiment

We conducted our field experiment at the University of Szeged (SZTE), which is the second-largest Hungarian university. The study program was initiated by the Directorate of Education of the university to develop a low-cost and easily scalable tool for decreasing dropout. The program was approved by the rector and senate of the university.

Our target population was those students engaged in full or correspondence-based education at SZTE, enrolled in the fall semester of the academic year 2019/2020, and attending classes taught in Hungarian (some students have classes taught in English). We only treated students in one study program (e.g., sociology) if they were involved in many programs (e.g., sociology and economics).

We preregistered 16,992 students at the university who met our inclusion criteria. After preregistration, 1,453 students (8.5%) changed their active status to passive; as we could not treat them, they were excluded from the analysis. Our target population therefore contained 15,539 students. The median age of the students was 22.2, and 57% were female.

Our sample size is powered to detect a Cohen's $d$ effect size of 0.03 with an 80% chance. Thus, the sample is large enough to detect even a substantially small effect.

## II. 3. The encouragement intervention

We treated students with an intervention that consisted of an encouragement message that students received before their end-of-semester exam. Treated students received an e-mail and an SMS (text) message. The email message consisted of the encouragement followed by an invitation to participate in the endline survey, for which a weblink was provided. The SMS message consisted of only the encouragement message without the weblink to the endline survey. The control students received an e-mail that consisted only of the invitation to participate in the endline survey, including the online survey weblink. Control students did not receive encouragement in the e-mail and did not receive an SMS message at all.

The English translation of the Hungarian text that treated students received in the e-mail message was as follows: "*Dear Student! The fact that you will soon take your exam proves that you already have many successful exams behind you! I truly hope that you will succeed in the next one as well, and I wish you every success! Please follow this weblink and answer three simple questions before your next exam. We will distribute vouchers worth a total of 100,000 HUF that can be redeemed at the SZTE Gift Shop among the respondents. Winners will be notified via e-mail. In the name of the Head of the Directorate of Education Péter Szakál.*" Our treatment message used a very similar sentence that Behncke [30] used successfully.

The first sentence of the e-mail message praises students for their prior achievements ("you already have many successful exams behind you"). The sentence confirms students' competence, and empowers them by pointing to their successes rather than their challenges. This sentence, therefore, is intended to raise students' self-efficacy as, according to Bandura [8], accomplishments of past performance and verbal persuasions are important sources of self-efficacy. The sentence also aims to influence students' test anxiety since positive affirmation messages decrease students' worries [31]. The sentence is valid for all students, since students have already taken successful exams to be admitted to the university.

The second sentence signals trust in students' success ("I truly hope that you will succeed"). The sentence is designed to be a self-fulfilling prophecy [41]. It is intended to affect students'

behavioral intention [15] by evoking their motivation to fulfill the meaning of the sentence [41, 42].

Students in the control group received an e-mail directing their attention to the endline survey and lottery without encouragement. They received the following message: "*Dear Student! Please follow this weblink to answer three simple questions before your next exam. We will distribute vouchers worth a total of 100,000 HUF that can be redeemed at the SZTE Gift Shop among the respondents. Winners will be notified via e-mail. In the name of the Head of the Directorate of Education Péter Szakál.*"

The sentence about the lottery in both the treated and control students' e-mails aimed to motivate students to fill in the endline questionnaire. The wording of the sentence prompted students to win vouchers by making a small effort and answering just three questions. In the SZTE gift shop, students could buy various products branded with the SZTE logo with the vouchers, like office supplies, mugs, t-shirts, sweatshirts, etc. The price of an average product is under 10,000 HUF (about 35 USD).

In addition to the e-mail message, we sent to treated students an SMS text message before their exam. Similar to the e-mail, the SMS messages contain the same elements (praise for past achievements and trust) in a more condensed form. The English translation of the Hungarian SMS sentence is as follows: "*We wish you good luck in your next exam since, during your educational career, you have already successfully proved your aptitude*! *SZTE Education Directorate*". Students in the control group did not receive any text messages on their mobile devices.

S8 Appendix contains the original Hungarian version of the e-mail and SMS encouragement messages.

We sent out the treated and control e-mail messages at 8 pm the day before the students' exam. The treatment SMS was sent out at 7 am on the day of the exams.

The motivations behind sending out the treatment message via two channels were threefold. First, our aim was to strengthen the treatment effect by sending out the encouragement twice, while varying the language and the channel of the message. Second, we aimed to encourage students relatively close to their exams, but we could only customize sending SMS messages (but not e-mails). Third, we aimed to collect endline data before students' exams. Nevertheless, students are unlikely to answer a questionnaire just before their exam. Therefore, only the e-mail contained the weblink to the online questionnaire.

We do not know precisely when, e.g., how long before the exam, students read the treatment messages. Nevertheless, the date when students filled in the endline survey indicates when they might have read the e-mail. Fig 1 shows when students completed the endline survey relative to the corresponding exam. On average, students filled in the questionnaire 13 hours before their exam. This means that students must have received the e-mail a couple of hours before their exam.

Fig 2 shows the time (in hours) relative to the exam when the treatment SMS was sent out to students' mobile devices. The majority of students (66%) received the treatment SMS within 3 hours before the exam, indicating that the SMS encouraged students shortly before their exams.

Fig 3 shows the numbers of total treatment messages (e-mails and SMS) that we sent out to students taking exams on the corresponding calendar date. Approximately 80% of treatment messages were sent out in the first ten days of the campaign. This indicates a condensed treatment period, mainly concentrated in the first few days of the exam period.

An online follow-up survey that we carried out after the treatment (for more details, see S1 Appendix) allowed us to speculate on the sources of treatment contamination. Our speculations focused on two points. First, the content of the treatment message might have been revealed to students before actually receiving it. Specifically, 33% of students shared the

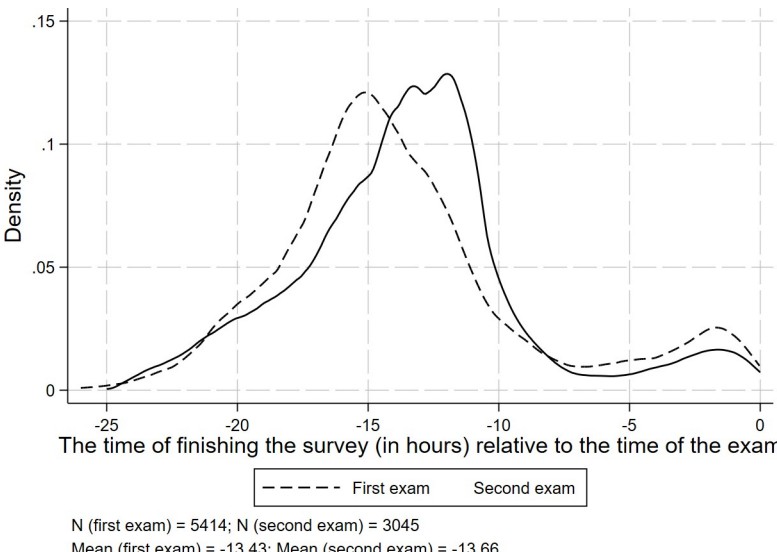

N (first exam) = 5414; N (second exam) = 3045
Mean (first exam) = -13.43; Mean (second exam) = -13.66

**Fig 1. The relative time difference in hours between finishing the survey and the beginning of the exam.**

message they received with university peers. By sharing the encouragement message, students attenuated the treatment effect, which made the estimations more conservative. Conceptually, our treatment, receiving the encouragement message, cannot be shared, and thus it is less exposed to the spillover effect.

Second, not receiving the encouragement message could discourage the untreated students and may lead to adverse treatment effects. We specifically asked students how sad they were when they found out that peers had received the encouragement message but they had not. Since, on a five-point Likert scale, 17% of students indicated that they were "sad" or "very sad" when they have not received the encouragement message, we conclude that the adverse treatment effect, which might have moved our estimation into an anticonservative direction, is moderate.

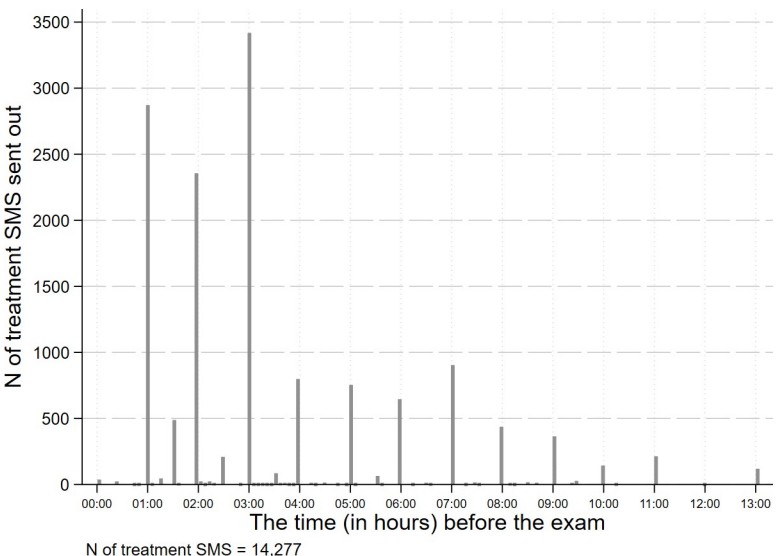

N of treatment SMS = 14,277

**Fig 2. The time (in hours) relative to the exam when the treatment SMS was sent.**

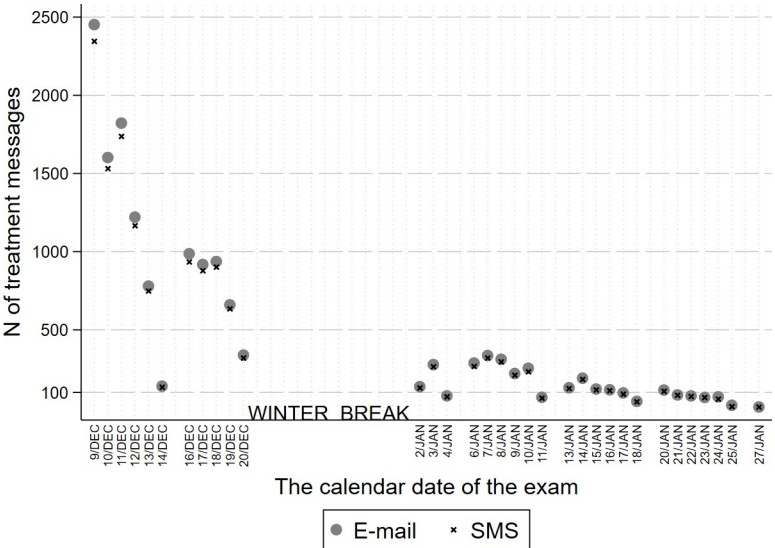

**Fig 3. The total number of treatment messages (e-mail and SMS) corresponding to an exam on a particular calendar date.** Note: E-mail messages were sent out at 8 pm the day before the exam. Text messages (SMS) were sent out at 7 am on the day of the exam. N of treatment e-mail = 14,974. N of treatment SMS = 14,277.

## II. 4. Study design and randomization

We designed a randomized field experiment with a crossover design in which students acted as their own control [43]. We randomized the ordering of the treatment (at the student level), i.e., when students received the treatment. We assigned all students to two consecutive conditions (treated/control), but we randomized the sequence of these conditions. Students randomized to *Group A* received the treatment before their first exam; in this case, the treatment condition preceded the control condition. Students randomized to *Group B* received the treatment before their second exam; in this case, the treatment condition followed the control condition.

Specifically, we allocated students to Group A/B based on pair-matched randomization [44]. First, we sorted the data file according to the following baseline variables: the study program in which the student is enrolled, the level of training, the type of training, the financial form of training, students' gender, and students' ability. In the sorted data file, students who followed each other were alike. We next identified the most similar two students: students who followed each other in the data file. In the next step, within each pair, we randomly assigned students to Group A or Group B based on the value of a randomly generated number.

In the analytic sample of students (N = 15,539), there are 7,771 students (50.01%) who were allocated to Group A and randomized to receive the encouragement message before the first exam. There are 7,768 students (49.99%) randomized to group B to receive the message before the second exam.

The design enabled us to observe all students under two conditions: when they received and when they did not receive the encouragement message. Note that we intended to treat both groups of students (A and B) for ethical reasons. Therefore, we intended to send each student two messages (one treatment and one control message).

We re-examined the treatment status after randomization at the end of the treatment period, when all messages had been sent out. We discovered that every student had received at least one e-mail message (before their first or second exam), but not every student had received the encouragement message (e.g., they only received the control message).

Students did not receive the treatment message if their teachers entered the exam in question in the university's registry after the exam had happened. In this case, we were not able to send students the encouragement message, since the corresponding exam was not listed in the university's registry at that time of treatment. In sum, 3.65% of students (N = 565) did not receive an encouragement message. Our analysis is, therefore, an intention-to-treat (ITT) analysis.

## II. 5. Balance test

Randomization resulted in groups that are well balanced with respect to the baseline covariates. Table 1 shows the differences in means between students allocated to Group A or Group B in each baseline covariate separately.

The mean difference between students in Group A (minus) those in Group B is quite small. There are only a few baseline variables (marked with bold) where the difference in means exceeds +/- 5 percentage points. Most notably, none of the differences between the two groups are statistically significant based on two-tailed t-tests.

## II. 6. Measures

**II. 6. 1. The outcome variables (Y).** The primary outcome variable is students' end-of-semester exam grades, measured in integers between 1 and 5. Grade 1 means fail. Other grades are equivalent to passing the exam, and in ascending order, they express the quality of students' performance, with 5 as the best. Relative grading is used in Hungary; that is, there is no absolute benchmark to which teachers relate students' performance.

**Table 1. Balance test.** The difference in means between students allocated to Group A relative to Group B for each baseline covariate separately.

|  | All students | Students with Endline Questionnaire | Students with Baseline Questionnaire |
|---|---|---|---|
| Female | -0.008 | 0.000 | -0.038 |
| Age | 0.000 | 0.001 | 0.001 |
| Students' ability | -0.002 | -0.002 | 0.001 |
| Students' ability is missing | -0.002 | 0.001 | 0.000 |
| Full-time training | 0.000 | -0.006 | 0.038 |
| State-financed training | 0.005 | 0.003 | -0.018 |
| Bachelor level | -0.001 | -0.002 | -0.004 |
| Master level | -0.003 | 0.001 | -0.014 |
| Undivided | 0.003 | 0.005 | 0.022 |
| Higher-level vocational training | 0.001 | -0.007 | -0.048 |
| First-year students | 0.003 | -0.005 | 0.009 |
| Exam difficulty | 0.001 | 0.015 | **0.074** |
| Exam difficulty is missing | 0.001 | -0.003 | 0.005 |
| Baseline test anxiety | 0.002[$] | -0.005[$ $] | 0.002 |
| Baseline self-confidence | -0.011[$] | -0.007[$ $] | -0.011 |
| External control | 0.019[$] | 0.003[$ $] | 0.019 |
| Parental education (university degree | 0.029[$] | 0.037[$ $] | 0.029 |
| N | 15,539 | 7,026 | -0.038 |

[*] The difference is significant at 5% level using a two-tailed t-test.

[$] N = 2,305

[$ $] N = 1,612.

**Bold** coefficients mark the mean differences that are larger than +/- 0.05.

In Hungary, like in many other countries, university students are required to take exams at the end of the semester. Exams can be either written or oral in nature. Students have to register for the exams on the university's online platform. They can change their registration up to 24 hours before an exam. Students who do not show up for an exam automatically fail unless a medical doctor certifies that the student was ill on the day of the exam. Therefore, the primary outcome has a missing value if a student did not show up to the exam and a medical doctor certified that he or she was ill. Missing values were not replaced.

The source of the primary outcome is the university's registry. We have information on the exam grades that students were awarded in a particular subject at a particular time and date.

The secondary outcome variables are self-efficacy (1), motivation (2), and test anxiety (3). We measured these outcomes with three single-item questions on a scale ranging from 0 to 10. The source of the secondary outcome variables is the endline questionnaire that treated and control students voluntarily answered before their exam. We preregistered to delete those answers that were answered after the corresponding exam. We deleted 2,940 answers since approximately 25% of the answers to the endline questionnaire were provided after the exam.

Fig 4 summarizes the questions we asked in the endline questionnaire and lists how the single-item measures correspond to the deployed secondary outcomes.

S5 Appendix lists the pairwise correlation between various psychological measures and the secondary outcome variables.

The S8 Appendix contains the original Hungarian version of the survey questions of the secondary outcome variables.

As students voluntarily answered the endline questionnaires, the secondary outcomes are available for a subsample of students. S2 Appendix summarizes the differences between the composition of various sub-samples with three highlights. First, in the subsample of those who answered the endline questionnaire, the share of students allocated to Group A versus Group B was the same as in the whole sample. Therefore, randomization was maintained with no differential selection between Groups A and B. Second, the treatment status significantly decreased students' willingness to answer the endline questionnaire; only directing students' attention to the lottery increased participation in the endline survey. Third, the subsample of students that filled in the survey was more advantaged. It contains younger and more able students who are more likely to be enrolled in full-time and state-financed education, and female students are also over-represented among them. Because the subsample of those with secondary outcomes is more advantaged, we warn against generalizing the results of the secondary outcomes to the entire analytic sample.

Since our primary outcome can take only five values, the chances to find significant treatment effects on students' exam grades are smaller than finding significant treatment effects on the secondary outcomes since these variables range between 0 and 10.

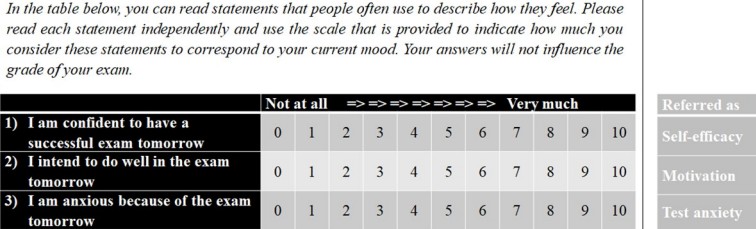

**Fig 4. Questions students answered before their first and second exam.**

Descriptive statistics of the outcome variables in the whole sample, and in the subsample of those who answered the endline questionnaire, are summarized in S3 Appendix.

**II. 6. 2. Treatment variable (T).**   The treatment variable (T) is a 0/1 variable that indicates whether the student received the encouragement message (T = 1), i.e., an e-mail and SMS before the exam. The treatment variable is coded as zero (T = 0) if students received the control message, which is an e-mail without encouragement, before their exam.

**II. 6. 3. The exam (E) and carry-over effects (T×E).**   Students' first and second exams were from different subjects, which may differ in format, scope, and difficulty. We captured these differences with a dummy variable (E), indicating whether the corresponding exam was a student's first (E = 0) or second exam E (= 1).

Students took their second exam soon after their first exam. The median student had four days between their first and second exams, and most frequently (in 21% of cases), there was only one day between the two exams.

The interaction of T and E indicates the carry-over effect, i.e., whether the ordering of the treatment influences the outcome variables. A significant carry-over effect biases the estimation of the average treatment effect [45].

In our design, we expect a negative carry-over effect, which means that encouraging students before their first exam affects their outcomes at the second exam. Since the sequence of treated and control conditions is either *treated-control* (Group A) or *control-treated* (Group B), treating students first might lead to a long-lasting effect or a long wash-out period. A statistically significant negative carry-over effect signals that the treatment effect is higher at students' first exam than at their second. A negative carry-over effect legitimizes the encouragement treatment and shows that students yearn for encouragement, since treating them before their first exam also affected their outcomes at the second exam, when they were not treated.

Under the current design, the carry-over effect does not provide a substantive interpretation of possible mechanisms that might lead to the longer-lasting effect when treating students before the first exam instead of the second.

**II. 6. 4. Control variables (X).**   The preregistered control variables and their coding are listed in S4 Appendix. The corresponding survey instruments are shown in the original Hungarian in S8 Appendix and in Englis in S9 Appendix.

**II. 6. 5. Variables exploring treatment heterogeneity (Z).**   Our analysis of treatment heterogeneity is exploratory and is not driven by particular theoretical considerations. We preregistered to explore treatment heterogeneity concerning the following baseline variables: self-confidence (1), students' ability (2), parental education (3), test anxiety (4), external control (5), students' status as a first-year student (6), students' gender as female (7), students' possession of a mobile phone number that was entered in the university's registry (8), the day (calculated from the beginning of the campaign) on which students received the message (9), and difficulty of the exam (10).

## III. Empirical analysis and hypothesis

### III. 1. Testing the main effects (Eq 1)

In our primary analysis, we hypothesize that receiving an encouragement message would increase students' end-of-semester exam grades.

To assess the treatment effect, we preregistered to use the following multilevel random-effects model:

$$Y_{ied} = \beta_0 + \beta_1 T_{ied} + \beta_2 E_{ied} + \beta_3 T_{ied} \times E_{ied} + \beta_4 X_{ied} + \varphi_{ied} + \mu_i + \epsilon_{ied} \qquad \text{(Eq 1)}$$

where $Y_{ied}$ is the *i*-th students' grade in exam *e* on day *d*. Variable T is the treatment (0/1). The

variable E refers students' second exam (first = 0/second = 1) and controls for differences between students' first and second exams. Variable X captures students' baseline variables measured before the treatment, obtained from the university's registry. We employ study-program-fixed-effect ($\varphi_{ied}$) and student-random-effect ($\mu_i$) effects.

In our secondary analysis, we substitute $Y_{ied}$ in Eq 1. with one of the corresponding secondary outcomes, i.e., on self-efficacy, motivation, and test anxiety, respectively.

The coefficients in Eq 1 are unstandardized regression coefficients. The coefficient $\beta_1$ identifies the causal treatment effect. It is the mean difference between treated and control students' outcomes concerning the first exam.

The coefficient $\beta_2$ identifies the period effect, i.e., the mean difference in control students' outcomes achieved at the second exam relative to the first. The coefficient does not have a causal interpretation, since the ordering of students' exams was not randomized. For example, a negative $\beta_2$ coefficient signals that students' outcomes are lower at the second exam. Differences in students' outcomes between the first and the second exam might be explained by various factors, including the difficulty of exams and students' fatigue and level of preparedness.

The coefficient $\beta_3$ identifies the carry-over effect—the difference in the treatment effect between the first and second exams. The coefficient is the difference of two mean-differences: The mean difference of outcomes between treated and control students in the second exam minus the mean difference of outcomes between treated and control students in the first exam.

The treatment effect relating to students' second exam is the linear combination of the coefficients $\beta_1$ and $\beta_3$.

Our hypothesis on the main treatment effect will be confirmed if we obtain a positive coefficient for $\beta_1$, and if we do not have a carry-over effect—i.e., if the main treatment effect concerning students' first and second exams do not differ statistically. We preregistered to use the 5% significance level concerning the primary outcome. Since we have three secondary outcomes in the secondary analyses, we preregistered here the family-wise error rate to deal with multiple testing errors [46, 47].

Specifically, we preregistered the following rules of decisions. We ordered p-values from low to high. With three secondary outcomes and the significance level of 0.05, the critical p-value would be 0.0167 for the coefficient with the lowest p-value (0.05* 1/3); this is the same as the Bonferroni correction. For the coefficient with the second-lowest p-value, the critical p-value would be 0.033 (0.05*2/3). For the coefficient with the highest p-value, the critical p-value would be 0.05 (0.05*3/3).

Alternative model specifications for calculating the main treatment effects are shown in S6 Appendix.

## III. 2. Testing treatment heterogeneity (Eq 2)

Hypotheses on treatment heterogeneity are exploratory. We hypothesized a greater treatment effect for students with: low self-confidence (1), a lower level of initial ability (2), and students whose parents do not have a university education (3).

We hypothesized a higher treatment effect for: anxious students (4), students with external control (5), first-year students (6), female students (7), students who had a phone number and thus received the text message parallel to the e-mail message (8), students who received the encouragement message at a later day calculated from the beginning of the campaign (9), and students who took a difficult exam (10).

In order to explore treatment heterogeneity, we included the preregistered baseline variables ($Z_{ied}$) in Eq 1. and in separate models, and we tested the two-way interaction of each of the Z variables with the treatment (T).

We estimated the following multilevel random-effects model to explore treatment heterogeneity:

$$Y_{ied} = \beta_0 + \beta_1 T_{ied} + \beta_2 E_{ied} + \beta_3 T_{ied} \times E_{ied} + \beta_4 X_{ied} + \beta_5 Z_{ied} + \beta_6 T_{ied} \times Z_{ied} + \varphi_{ied} + \mu_i$$
$$+ \epsilon_{ied} \quad \text{(Eq 2)}$$

In Eq 2 the coefficient $\beta_6$ shows the treatment heterogeneity.

### III. 3. The preregistered mediation analysis

We preregistered a mediation analysis that aimed to explore the mechanism through which the encouragement message influences exam grades. Since the main treatment effect concerning students' exam grades was not significant in any subsamples, we do not show the mediation analysis results in the paper. The results of the preregistered mediation analysis are, however, available in the S10 Appendix.

## IV. Results

### IV. 1. Bivariate raw results

Fig 5 visualizes the unconditional raw mean of primary and secondary outcome variables in treated and control groups with 95% confidence intervals.

It is notable that in the case of three of the four outcome variables, the mean values are slightly above the theoretical middle point of the measurement scales' range. Students' motivation is the only outcome variable where the means are close to the theoretical maximum of the measurement scale, suggesting that all students were highly motivated. Thus, the potential to change students' motivation by a light-touch intervention might be limited.

The differences between the means of treated and control students are statistically significant in the case of self-efficacy ($p < 0.005$) and motivation ($p = 0.032$) and are not statistically significant in the case of exam grades and test anxiety. All the differences are quite small. Our multivariate analyses will go behind these raw differences.

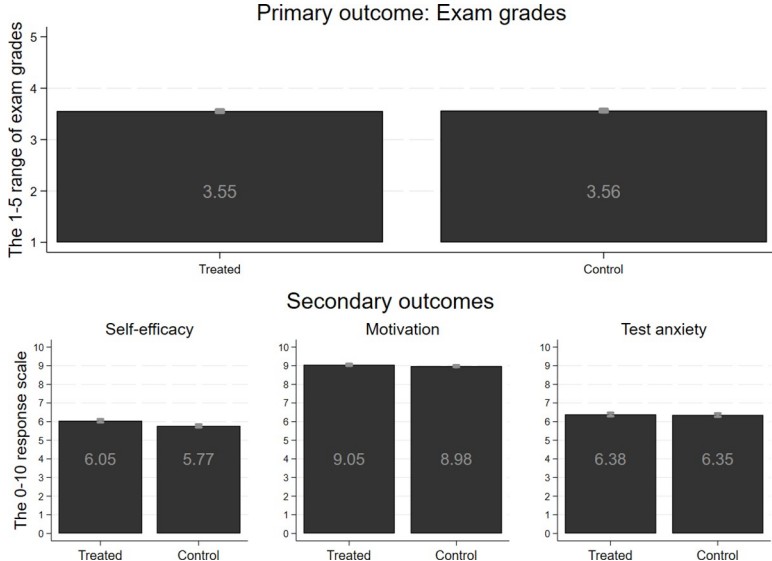

**Fig 5. The unconditional raw mean of primary and secondary outcome variables in treated and control groups with 95% confidence intervals.**

## IV. 2. The results of the multivariate analyses

We present the results of our multivariate analyses in tables, in which the first row indicates the treatment effect concerning students' first exam ($\beta_1$), while the second row reveals the outcome differences between the first and second exams ($\beta_2$). The third row of the table shows the difference in treatment effect between students' first and second exams ($\beta_3$). The treatment effect concerning students' second exam ($\beta_1+\beta_3$) is indicated in the last row of the tables.

Column 1 shows the main treatment effect, while Columns 2–7 summarize the interaction effects with various preregistered baseline variables obtained from the university's register. Column 8 shows the main treatment effect in the restricted sample. Since we collected baseline variables with a baseline survey, some preregistered baseline variables are available for a restrictive sample of those who answered the baseline questionnaire. Columns 9–12 summarize the interaction effects with the baseline variables collected with the baseline survey.

**IV. 2. 1. Exam grades.** Table 2 summarizes the results for the exam grades. The first row of Column 1 shows that students who received the encouragement message did not gain higher exam grades at their first exam ($\beta_1 = 0.017$; p = 0.418). The Cohen's *d* effect size, which expresses the treatment effect in standard deviation units of the outcome variable, is small (0.011).

Students performed worse in their second exam ($\beta_2 = -0.075$; p < 0.001) than in their first exam. The results show no carry-over effect ($\beta_3 = -0.040$; p = 0.208). Thus, the treatment effect was similar at students' first and second exams. In other words, receiving the encouragement message before the first exam did not have an enduring effect on students' exam grades.

As Column 2 indicates, we explored treatment heterogeneity in students' baseline ability ($\beta_6 = 0.033$; p = 0.040). More able students gained a larger increase in their grades. Since we hypothesized that students with lower ability would gain more from the treatment, the result contradicts our preregistered hypothesis.

Fig 6 shows the treatment heterogeneity based on students' baseline ability. For example, among those students whose baseline ability was one standard deviation higher than the average, the encouragement message induced an increase (coef. = 0.049; p = 0.056) in their exam grades, which is statistically marginally significant.

We did not find any other treatment heterogeneity regarding students' exam grades.

**IV. 2. 2. Self-efficacy.** Column 1 in Table 3 experimentally confirms a significant positive treatment effect on students' self-efficacy. Receiving the encouragement message increased students' self-efficacy by $\beta_1 = 0.304$ (p < 0.001) unit, which is a Cohen's d effect size of 0.12. We preregistered to use the significance level of 0.0167 to correct for multiple testing in the secondary outcomes. Since the corresponding p-value is 0.00000322319, the estimated coefficient is highly significant at the preregistered level.

Students reported less self-efficacy before their second exam ($\beta_2 = -0.193$; p = 0.008). The treatment effect did not differ between students' first and second exams, ($\beta_3 = 0.027$; p = 0.818), which suggests there was no carry-over effect. Therefore, the treatment had the same effect on students' self-efficacy when delivered before their first or their second exams. Specifically, as the last row of Column 1 in Table 3 shows, the joint linear effect of $\beta_1$ and $\beta_3$ (the treatment effect concerning students' second exam) is 0.277; the coefficient is significant at the 1% significance level.

Compared to the full sample (Column 1), the main treatment effect was somewhat lower (Column 8; $\beta_1 = 0.285$; p = 0.031) in the sample of those who have baseline survey data. The difference in the treatment effect between the full and restricted samples (e.g., between Column 1 and Column 8) was not statistically significant (p = 0.860). There was no carry-over effect in the restricted sample (Column 8; $\beta_3 = -0.033$; p = 0.894). The treatment effect

**Table 2. Treatment effect on students' endline exam grades, unstandardized regression coefficients.**

| | (1) | (2) | (3) | (4) | (5) | (6) | (7) | (8) | (9) | (10) | (11) | (12) |
|---|---|---|---|---|---|---|---|---|---|---|---|---|
| $\beta_1$: Treated [T] | 0.017 | 0.016 | 0.011 | 0.039 | -0.006 | 0.018 | 0.003 | 0.048 | 0.049 | 0.052 | 0.051 | 0.031 |
| (treated = 1) | (0.021) | (0.021) | (0.022) | (0.026) | (0.065) | (0.022) | (0.023) | (0.052) | (0.052) | (0.052) | (0.052) | (0.064) |
| $\beta_2$: Exam [E] (second = 1) | -0.075*** | -0.075*** | -0.075*** | -0.075*** | -0.075*** | 0.052* | -0.074*** | -0.025 | -0.023 | -0.021 | -0.022 | -0.026 |
| | (0.021) | (0.021) | (0.021) | (0.021) | (0.021) | (0.022) | (0.021) | (0.053) | (0.053) | (0.053) | (0.053) | (0.053) |
| $\beta_3$: Carry-over [T×E] | -0.040 | -0.040 | -0.040 | -0.040 | -0.040 | -0.038 | -0.043 | -0.079 | -0.082 | -0.085 | -0.086 | -0.078 |
| | (0.032) | (0.032) | (0.032) | (0.032) | (0.032) | (0.032) | (0.032) | (0.080) | (0.080) | (0.079) | (0.080) | (0.080) |
| $\beta_6$: Interaction[a] | | 0.033* | 0.019 | -0.039 | 0.024 | -0.000 | 0.083 | | 0.007 | -0.009 | 0.017 | 0.031 |
| (T×Main effet[Z]) | | (0.016) | (0.029) | (0.028) | (0.064) | (0.001) | (0.058) | | (0.035) | (0.034) | (0.035) | (0.069) |
| $\beta_5$: Main effects[Z] | | | | | | | | | | | | |
| Baseline test anxiety[b] | | | | | | | | ✓ | -0.080** | ✓ | ✓ | ✓ |
| | | | | | | | | | (0.027) | | | |
| Baseline self-confidence[b] | | | | | | | | ✓ | ✓ | 0.168*** | ✓ | ✓ |
| | | | | | | | | | | (0.027) | | |
| Baseline external control[b] | | | | | | | | ✓ | ✓ | | -0.072** | ✓ |
| | | | | | | | | | | | (0.027) | |
| Parental education | | | | | | | | ✓ | ✓ | ✓ | ✓ | 0.009 |
| | | | | | | | | | | | | (0.055) |
| Students' ability[b] | ✓ | 0.208*** | ✓ | ✓ | ✓ | ✓ | ✓ | ✓ | ✓ | ✓ | ✓ | ✓ |
| | | (0.015) | | | | | | | | | | |
| First-year student | ✓ | ✓ | -0.102*** | ✓ | ✓ | ✓ | ✓ | ✓ | ✓ | ✓ | ✓ | ✓ |
| | | | (0.025) | | | | | | | | | |
| Female | ✓ | ✓ | ✓ | 0.177*** | ✓ | ✓ | ✓ | ✓ | ✓ | ✓ | ✓ | ✓ |
| | | | | (0.023) | | | | | | | | |
| Has mobile phone | | | | | -0.005 | | | | | | | |
| | | | | | (0.050) | | | | | | | |
| Day of message | | | | | | -0.019*** | | | | | | |
| | | | | | | (0.001) | | | | | | |
| Exam difficulty | ✓ | ✓ | ✓ | ✓ | ✓ | ✓ | -1.659*** | ✓ | ✓ | ✓ | ✓ | ✓ |
| | | | | | | | (0.047) | | | | | |
| **Constant** | 3.730*** | 3.735*** | 3.732*** | 3.718*** | 3.735*** | 3.550*** | 3.733*** | 2.305* | 2.323* | 2.348* | 2.372* | 2.277* |
| | (0.354) | (0.354) | (0.354) | (0.354) | (0.357) | (0.348) | (0.354) | (1.081) | (1.079) | (1.070) | (1.080) | (1.082) |
| Observations | 28,156 | 28,156 | 28,156 | 28,156 | 28,156 | 28,156 | 28,156 | 4,335 | 4,335 | 4,335 | 4,335 | 4,335 |
| N of students | 15,264 | 15,264 | 15,264 | 15,264 | 15,264 | 15,264 | 15,264 | 2,295 | 2,295 | 2,295 | 2,295 | 2,295 |
| Cohen's $d$ effect size of $\beta_1$ | 0.011 | 0.011 | 0.007 | 0.027 | -0.004 | 0.012 | 0.002 | 0.034 | 0.035 | 0.037 | 0.037 | 0.022 |
| The joint linear effect of $\beta_1$ & $\beta_3$ | -0.023 | -0.024 | -0.029 | -0.000 | -0.046 | -0.020 | -0.040 | -0.031 | -0.033 | -0.034 | -0.035 | -0.047 |
| | (0.021) | (0.021) | (0.023) | (0.027) | (0.065) | (0.026) | (0.025) | (0.053) | (0.053) | (0.053) | (0.053) | (0.065) |

All models (Column 1–12) contain the following preregistered standard baseline control variables: student's gender, age, ability, student is a first-year student, the type of training, the financial form of training, the level of training, the difficulty of the exam, and study program fixed effects.

The table lists those variables that we preregistered as a variable to test treatment heterogeneity (Z). Some of the standard control variables are listed in the table as they appear among variables in Z. We marked these variables with the ✓ sign indicating that the given variable was included in the regression even though its estimated coefficient was not included in the table.

In addition to the standard baseline variables, columns 8–12 contain the following preregistered additional baseline variables from the baseline survey, and thus they are available for a subset of students: baseline test anxiety, baseline self-confidence, baseline external control, and parental education. Since all of the additionally used control variables were preregistered as a variable to test treatment heterogeneity (Z), all of them are listed in the table and therefore marked with the ✓ sign.

[a] To enhance readability, the *Interaction* (T×Z) refers to the product of the treatment variable (T) and a specific main effect (Z). The coefficient of the corresponding main effect is shown in the table. For example, in Column 2, the interaction refers to the product of T×Students' ability, and in Column 10, the *Interaction* refers to the product of T× Baseline self-confidence.

[b] z-standardized variable at 0 mean and 1 standard deviation.

Standard errors in parentheses

*** p<0.001

** p<0.01

* p<0.05, + p<0.1.

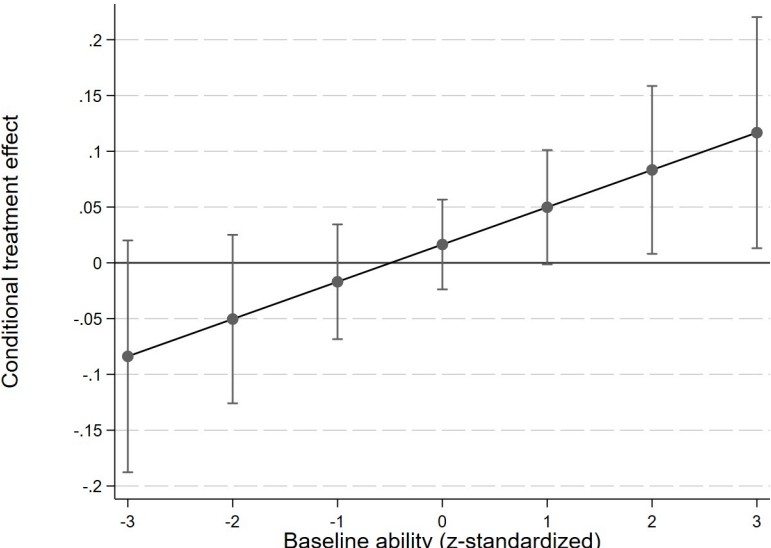

**Fig 6. Conditional treatment effect of receiving the encouragement message on students' endline exam grades, based on students' baseline ability.**

concerning students' second exam was, however, statistically not significant ($\beta_1+\beta_3 = 0.253$; p = 0.148), most likely due to the smaller sample size, which increased the standard errors of the estimations.

There is no treatment heterogeneity in the full sample (Columns 2–7). In the restricted sample of those who answered the baseline survey, however, the encouragement message increased anxious students' self-efficacy (Column 9; $\beta_6 = 0.227$; p = 0.011) and also the self-efficacy of those students' whose baseline self-confidence was low (Column 10; $\beta_6 = -0.171$; p = 0.051).

**IV. 2. 3. Motivation.** Column 1 in Table 4 shows how the encouragement message influenced students' motivation to do well in the exam. We have experimentally confirmed that those who received the encouragement message experienced a 0.101 unit increase in their motivation (p = 0.013), equivalent to a Cohen's *d* effect size of 0.066. We preregistered to use the significance level of 0.033 to correct for multiple testing in the secondary outcomes. Since the corresponding p-value is below this threshold, the estimated coefficient is significant at the preregistered level.

Results show no difference in students' self-reported motivation ($\beta_2 = 0.000$, p = 0.995) between the first and second exam.

The marginally significant and negative carry-over effect ($\beta_3 = -0.121$ p = 0.093) shows that the treatment affected students' motivation to a smaller extent at their second exam than at the first exam. Even though the carry-over effect is only marginally significant, we suggest a cautious interpretation of the treatment effect since the encouragement message did not affect students' motivation before their second exam (0.101 + (-0.121) = -0.020, p = 0.703); the encouragement only affected students' first exam and was not replicated in the second exam.

Compared to the full sample, the treatment effect is estimated to be smaller in the restricted sample among those who filled in the baseline background questionnaire. The difference between the effects (Column 1 and Column 8) is not statistically significant (-0.045; p = 0.506).

As shown in Column 3, the treatment had a larger effect on older students ($\beta_6 = 0.149$; p = 0.001) and had no impact for first-year students (0.149 + (-0.151) = -0.001, p = 0.977). These findings contradict our hypothesis that first-year students, who were actually taking

**Table 3. Treatment effect on students' endline self-efficacy, unstandardized regression coefficients.**

| | (1) | (2) | (3) | (4) | (5) | (6) | (7) | (8) | (9) | (10) | (11) | (12) |
|---|---|---|---|---|---|---|---|---|---|---|---|---|
| $\beta_1$: **Treated [T]** | 0.304*** | 0.306*** | 0.344*** | 0.316*** | 0.628** | 0.291*** | 0.233** | 0.285* | 0.289* | 0.329** | 0.277* | 0.231 |
| *(treated = 1)* | (0.065) | (0.065) | (0.073) | (0.087) | (0.219) | (0.070) | (0.075) | (0.132) | (0.127) | (0.123) | (0.131) | (0.166) |
| $\beta_2$: **Exam [E]** *(second = 1)* | -0.193** | -0.194** | -0.194** | -0.193** | -0.192** | 0.000 | -0.181* | -0.089 | -0.063 | -0.009 | -0.091 | -0.085 |
| | (0.073) | (0.073) | (0.073) | (0.073) | (0.073) | (0.077) | (0.073) | (0.151) | (0.146) | (0.142) | (0.150) | (0.151) |
| $\beta_3$: **Carry-over [T×E]** | -0.027 | -0.026 | -0.023 | -0.027 | -0.027 | -0.016 | -0.048 | -0.033 | -0.013 | -0.105 | -0.024 | -0.041 |
| | (0.116) | (0.116) | (0.116) | (0.116) | (0.116) | (0.121) | (0.117) | (0.246) | (0.235) | (0.226) | (0.244) | (0.246) |
| $\beta_6$: **Interaction**[a] | | -0.057 | -0.125 | -0.021 | -0.341 | 0.001 | 0.385+ | | 0.227* | -0.171+ | 0.084 | 0.111 |
| (T×Main effet[Z]) | | (0.056) | (0.100) | (0.096) | (0.219) | (0.004) | (0.200) | | (0.089) | (0.088) | (0.091) | (0.182) |
| $\beta_5$:**Main effects[Z]** | | | | | | | | | | | | |
| Baseline test anxiety[b] | | | | | | | | ✓ | -0.828*** | ✓ | ✓ | ✓ |
| | | | | | | | | | (0.073) | | | |
| Baseline self-confidence[b] | | | | | | | | ✓ | ✓ | 1.085*** | ✓ | ✓ |
| | | | | | | | | | | (0.070) | | |
| Baseline external control[b] | | | | | | | | ✓ | ✓ | ✓ | -0.333*** | ✓ |
| | | | | | | | | | | | (0.073) | |
| Parental education | | | | | | | | ✓ | ✓ | ✓ | ✓ | -0.210 |
| | | | | | | | | | | | | (0.152) |
| Students' ability[b] | ✓ | 0.104* | ✓ | ✓ | ✓ | ✓ | ✓ | ✓ | ✓ | ✓ | ✓ | ✓ |
| | | (0.050) | | | | | | | | | | |
| First-year student | ✓ | ✓ | 0.157+ | ✓ | ✓ | ✓ | ✓ | ✓ | ✓ | ✓ | ✓ | ✓ |
| | | | (0.082) | | | | | | | | | |
| Female | ✓ | ✓ | ✓ | -0.397*** | ✓ | ✓ | ✓ | ✓ | ✓ | ✓ | ✓ | ✓ |
| | | | | (0.079) | | | | | | | | |
| Has mobile phone | | | | | 0.169 | | | | | | | |
| | | | | | (0.173) | | | | | | | |
| Day of message | | | | | | -0.024*** | | | | | | |
| | | | | | | (0.003) | | | | | | |
| Exam difficulty | ✓ | ✓ | ✓ | ✓ | ✓ | ✓ | -1.621*** | ✓ | ✓ | ✓ | ✓ | ✓ |
| | | | | | | | (0.156) | | | | | |
| **Constant** | 7.612*** | 7.600*** | 7.597*** | 7.610*** | 7.430*** | 7.422*** | 7.641*** | 6.728* | 7.551** | 6.522* | 6.995* | 6.872* |
| | (1.401) | (1.401) | (1.401) | (1.401) | (1.409) | (1.394) | (1.400) | (2.977) | (2.848) | (2.738) | (2.957) | (2.979) |
| Observations | 8,296 | 8,296 | 8,296 | 8,296 | 8,296 | 8,296 | 8,296 | 2,016 | 2,016 | 2,016 | 2,016 | 2,016 |
| N of students | 6,908 | 6,908 | 6,908 | 6,908 | 6,908 | 6,908 | 6,908 | 1,594 | 1,594 | 1,594 | 1,594 | 1,594 |
| Cohen's $d$ effect size of $\beta_1$ | 0.120 | 0.121 | 0.136 | 0.125 | 0.249 | 0.115 | 0.092 | 0.111 | 0.112 | 0.128 | 0.108 | 0.090 |
| The joint linear effect of $\beta_1$ & $\beta_3$ | 0.277** | 0.280** | 0.321** | 0.289** | 0.601** | 0.275** | 0.185+ | 0.253 | 0.275 | 0.224 | 0.253 | 0.190 |
| | (0.085) | (0.085) | (0.092) | (0.102) | (0.226) | (0.106) | (0.098) | (0.175) | (0.170) | (0.164) | (0.174) | (0.201) |

All models (Column 1–12) contain the following preregistered standard baseline control variables: student's gender, age, ability, student is a first-year student, the type of training, the financial form of training, the level of training, the difficulty of the exam, and study program fixed effects.

The table lists those variables that we preregistered as a variable to test treatment heterogeneity (Z). Some of the standard control variables are listed in the table as they appear among variables in Z. We marked these variables with the ✓ sign indicating that the given variable was included in the regression even though its estimated coefficient was not included in the table.

In addition to the standard baseline variables, columns 8–12 contain the following preregistered additional baseline variables from the baseline survey, and thus they are available for a subset of students: baseline test anxiety, baseline self-confidence, baseline external control, and parental education. Since all of the additionally used control variables were preregistered as a variable to test treatment heterogeneity (Z), all of them are listed in the table and therefore marked with the ✓ sign.

[a] To enhance readability, the *Interaction* (T×Z) refers to the product of the treatment variable (T) and a specific main effect (Z). The coefficient of the corresponding main effect is shown in the table. For example, in Column 2, the interaction refers to the product of T×Students' ability, and in Column 10, the *Interaction* refers to the product of T× Baseline self-confidence.

[b] z-standardized variable at 0 mean and 1 standard deviation.

Standard errors in parentheses

*** p<0.001

** p<0.01

* p<0.05

+ p<0.1.

**Table 4. Treatment effect on students' endline motivation, unstandardized regression coefficients.**

| | (1) | (2) | (3) | (4) | (5) | (6) | (7) | (8) | (9) | (10) | (11) | (12) |
|---|---|---|---|---|---|---|---|---|---|---|---|---|
| $\beta_1$: **Treated [T]** | 0.101* | 0.100* | 0.149*** | 0.100+ | 0.229+ | 0.100* | 0.075 | 0.052 | 0.052 | 0.057 | 0.051 | 0.170+ |
| *(treated = 1)* | (0.041) | (0.041) | (0.045) | (0.054) | (0.138) | (0.044) | (0.047) | (0.078) | (0.078) | (0.078) | (0.078) | (0.098) |
| $\beta_2$: **Exam [E]** *(second = 1)* | -0.000 | -0.000 | -0.002 | -0.000 | 0.000 | 0.046 | 0.004 | 0.028 | 0.027 | 0.037 | 0.028 | 0.029 |
| | (0.045) | (0.045) | (0.045) | (0.045) | (0.045) | (0.048) | (0.046) | (0.090) | (0.090) | (0.090) | (0.090) | (0.089) |
| $\beta_3$: **Carry-over [T×E]** | -0.121+ | -0.121+ | -0.116 | -0.121+ | -0.120+ | -0.116 | -0.128+ | -0.121 | -0.123 | -0.130 | -0.121 | -0.129 |
| | (0.072) | (0.072) | (0.072) | (0.072) | (0.072) | (0.076) | (0.073) | (0.144) | (0.144) | (0.144) | (0.144) | (0.144) |
| $\beta_6$: **Interaction**[a] | | 0.009 | -0.151* | 0.001 | -0.135 | -0.000 | 0.138 | | 0.040 | -0.004 | 0.008 | -0.203+ |
| (T×Main effet[Z]) | | (0.035) | (0.063) | (0.060) | (0.139) | (0.003) | (0.125) | | (0.055) | (0.055) | (0.055) | (0.109) |
| $\beta_5$: **Main effects[Z]** | | | | | | | | | | | | |
| Baseline test anxiety[b] | | | | | | | | ✓ | 0.005 | ✓ | ✓ | ✓ |
| | | | | | | | | | (0.045) | | | |
| Baseline self-confidence[b] | | | | | | | | ✓ | ✓ | 0.110* | ✓ | ✓ |
| | | | | | | | | | | (0.045) | | |
| Baseline external control[b] | | | | | | | | ✓ | ✓ | ✓ | -0.053 | ✓ |
| | | | | | | | | | | | (0.043) | |
| Parental education | | | | | | | | ✓ | ✓ | ✓ | ✓ | -0.069 |
| | | | | | | | | | | | | (0.090) |
| Students' ability[b] | ✓ | 0.001 | ✓ | ✓ | ✓ | ✓ | ✓ | ✓ | ✓ | ✓ | ✓ | ✓ |
| | | (0.031) | | | | | | | | | | |
| First-year student | ✓ | ✓ | 0.010 | ✓ | ✓ | ✓ | ✓ | ✓ | ✓ | ✓ | ✓ | ✓ |
| | | | (0.051) | | | | | | | | | |
| Female | ✓ | ✓ | ✓ | 0.150** | ✓ | ✓ | ✓ | ✓ | ✓ | ✓ | ✓ | ✓ |
| | | | | (0.049) | | | | | | | | |
| Has mobile phone | ✓ | ✓ | ✓ | ✓ | -0.044 | | | | | | | |
| | | | | | (0.054) | | | | | | | |
| Day of message | ✓ | ✓ | ✓ | ✓ | ✓ | -0.006** | | | | | | |
| | | | | | | (0.002) | | | | | | |
| Exam difficulty | ✓ | ✓ | ✓ | ✓ | ✓ | ✓ | -0.177+ | ✓ | ✓ | ✓ | ✓ | ✓ |
| | | | | | | | (0.098) | | | | | |
| **Constant** | 9.206*** | 9.208*** | 9.170*** | 9.206*** | 9.155*** | 9.132*** | 9.213*** | 15.151*** | 15.265*** | 14.807*** | 15.190*** | 15.037*** |
| | (0.973) | (0.973) | (0.973) | (0.973) | (0.978) | (0.973) | (0.973) | (1.944) | (1.948) | (1.944) | (1.944) | (1.942) |
| Observations | 8,301 | 8,301 | 8,301 | 8,301 | 8,301 | 8,301 | 8,301 | 2,016 | 2,016 | 2,016 | 2,016 | 2,016 |
| N of students | 6,916 | 6,916 | 6,916 | 6,916 | 6,916 | 6,916 | 6,916 | 1,592 | 1,592 | 1,592 | 1,592 | 1,592 |
| Cohen's $d$ effect size of $\beta_1$ | 0.066 | 0.066 | 0.098 | 0.065 | 0.149 | 0.065 | 0.049 | 0.035 | 0.035 | 0.038 | 0.034 | 0.114 |
| The joint linear effect of $\beta_1$ & $\beta_3$ | -0.020 | -0.021 | 0.033 | -0.021 | 0.108 | -0.016 | -0.053 | -0.069 | -0.071 | -0.073 | -0.070 | 0.041 |
| | (0.053) | (0.053) | (0.058) | (0.064) | (0.143) | (0.066) | (0.061) | (0.104) | (0.104) | (0.104) | (0.104) | (0.120) |

All models (Column 1–12) contain the following preregistered standard baseline control variables: student's gender, age, ability, student is s first-year student, the type of training, the financial form of training, the level of training, the difficulty of the exam, and study program fixed effects.

The table lists those variables that we preregistered as a variable to test treatment heterogeneity (Z). Some of the standard control variables are listed in the table as they appear among variables in Z. We marked these variables with the ✓ sign indicating that the given variable was included in the regression even though its estimated coefficient was not included in the table.

In addition to the standard baseline variables, columns 8–12 contain the following preregistered additional baseline variables from the baseline survey, and thus they are available for a subset of students: baseline test anxiety, baseline self-confidence, baseline external control, and parental education. Since all of the additionally used control variables were preregistered as a variable to test treatment heterogeneity (Z), all of them are listed in the table and therefore marked with the ✓ sign.

[a] To enhance readability, the *Interaction* (T×Z) refers to the product of the treatment variable (T) and a specific main effect (Z). The coefficient of the corresponding main effect is shown in the table. For example, in Column 2, the interaction refers to the product of T×Students' ability, and in Column 10, the *Interaction* refers to the product of T× Baseline self-confidence.

[b] z-standardized variable at 0 mean and 1 standard deviation.

Standard errors in parentheses

*** p<0.001

** p<0.01

* p<0.05

+ p<0.1.

their first university exam and thus lacked prior experience with university exams, would gain more benefit from the encouragement campaign. The results indicate, however, that those older students who have possibly acquired a set of good/bad exam experiences are those who need encouragement to spur their motivation.

**IV. 2. 4. Test anxiety.** Table 5 (Column 1) shows that there was no treatment effect on students' test anxiety ($\beta_1$ = -0.053; p = 0.480) concerning their first exam. Those students who received encouragement messages reported lower test anxiety than students in the control group who did not receive encouragement messages. The differences are not, however, statistically significant.

Students reported less test anxiety before their second exam than they did before their first exam. Differences in test anxiety between students' first and second exams are not, however, statistically significant ($\beta_2$ = -0.072; p = 0.387).

The carry-over effect is statistically not significant ($\beta_3$ = 0.092; p = 0.492). Thus, the ordering of the treatment does not generate differences in students' test anxiety after.

The main effect ($\beta_1$ = -0.161; p = 0.297) of the treatment is somewhat larger (more negative) in the restricted sample (among those students who filled in the baseline questionnaire (Column 8). The difference in the treatment effect between the full and restricted samples is statistically not significant (p = 0.994).

The treatment effect increases (becomes more negative) during the intervention period, as indicated by the negative interaction coefficient in Column 6. With each day that is spent relative to the beginning of the treatment period, the (negative) effect increases by $\beta_6$ = -0.01 (p = 0.036). Thus, receiving the encouragement message decreases students' test anxiety significantly from the middle of the treatment period.

As hypothesized, the treatment decreased test anxiety of students with average (or below average) self-confidence (Column 10). Since the interaction coefficient is positive ($\beta_6$ = 0.261 p = 0.014), if students' baseline self-confidence increases, the (otherwise negative) treatment effect gradually diminishes. Among students with high self-confidence, the treatment has, however, no effect.

## IV. 3. Summary of treatment heterogeneity

We summarized the preregistered hypothesis about treatment heterogeneity in Table 6. Most of the hypotheses were not supported since the corresponding coefficient was not statistically significant (marked with NS in the table).

We found treatment heterogeneity by students' baseline ability in our primary outcomes but did not explore any other heterogeneous effect concerning students' exam grades. The result is exploratory due to multiple testing.

Significant interaction coefficients occurred sporadically across the three secondary outcome variables without a systematic pattern. Since the numbers of performed tests were large, significant interaction coefficients might have occurred by chance due to multiple testing. In other words, our results on treatment heterogeneity are exploratory, and future experimental research should confirm our exploratory results.

In the secondary outcomes, we can at best claim treatment heterogeneity according to students' baseline self-confidence. We found significant interaction coefficients for two out of the three secondary outcome variables (test anxiety and self-efficacy), as shown in Fig 7.

## V. Discussion

We carried out a large-scale, preregistered, randomized field experiment at the University of Szeged in Hungary (N = 15,539 students). We tested the impact of a light-touch automated

**Table 5. Treatment effect on students' endline test anxiety, unstandardized regression coefficients.**

| | (1) | (2) | (3) | (4) | (5) | (6) | (7) | (8) | (9) | (10) | (11) | (12) |
|---|---|---|---|---|---|---|---|---|---|---|---|---|
| $\beta_1$: Treated [T] | -0.053 | -0.053 | 0.003 | -0.026 | -0.387 | 0.014 | -0.049 | -0.161 | -0.173 | -0.202 | -0.160 | -0.320+ |
| (treated = 1) | (0.075) | (0.075) | (0.084) | (0.100) | (0.251) | (0.081) | (0.087) | (0.155) | (0.142) | (0.151) | (0.155) | (0.194) |
| $\beta_2$: Exam [E] (second = 1) | -0.073 | -0.073 | -0.074 | -0.072 | -0.074 | -0.171+ | -0.073 | -0.211 | -0.279+ | -0.291+ | -0.209 | -0.208 |
| | (0.084) | (0.084) | (0.084) | (0.084) | (0.084) | (0.089) | (0.084) | (0.177) | (0.164) | (0.173) | (0.177) | (0.177) |
| $\beta_3$: Carry-over [T×E] | 0.092 | 0.093 | 0.099 | 0.092 | 0.092 | 0.175 | 0.094 | 0.352 | 0.334 | 0.425 | 0.349 | 0.349 |
| | (0.135) | (0.135) | (0.135) | (0.135) | (0.135) | (0.141) | (0.135) | (0.287) | (0.259) | (0.279) | (0.287) | (0.287) |
| $\beta_6$: Interaction[a] | | -0.013 | -0.174 | -0.045 | 0.351 | -0.011* | -0.023 | | -0.083 | 0.261* | -0.076 | 0.293 |
| (T×Main effet[Z]) | | (0.065) | (0.114) | (0.110) | (0.252) | (0.005) | (0.230) | | (0.103) | (0.106) | (0.107) | (0.214) |
| $\beta_5$: Main effects[Z] | | | | | | | | | | | | |
| Baseline test anxiety[b] | | | | | | | | ✓ | 1.307*** | ✓ | ✓ | ✓ |
| | | | | | | | | | (0.081) | | | |
| Baseline self-confidence[b] | | | | | | | | ✓ | ✓ | -0.808*** | ✓ | ✓ |
| | | | | | | | | | | (0.086) | | |
| Baseline external control[b] | | | | | | | | ✓ | ✓ | ✓ | 0.080 | ✓ |
| | | | | | | | | | | | (0.086) | |
| Parental education | | | | | | | | ✓ | ✓ | ✓ | ✓ | -0.223 |
| | | | | | | | | | | | | (0.178) |
| Students' ability[b] | ✓ | -0.049 | ✓ | ✓ | ✓ | ✓ | ✓ | ✓ | ✓ | ✓ | ✓ | ✓ |
| | | (0.057) | | | | | | | | | | |
| First-year student | ✓ | ✓ | 0.129 | ✓ | ✓ | ✓ | ✓ | ✓ | ✓ | ✓ | ✓ | ✓ |
| | | | (0.094) | | | | | | | | | |
| Female | ✓ | ✓ | ✓ | 1.120*** | ✓ | ✓ | ✓ | ✓ | ✓ | ✓ | ✓ | ✓ |
| | | | | (0.091) | | | | | | | | |
| Has mobile phone | | | | | -0.156 | | | | | | | |
| | | | | | (0.200) | | | | | | | |
| Day of message | | | | | | 0.012** | | | | | | |
| | | | | | | (0.004) | | | | | | |
| Exam difficulty | ✓ | ✓ | ✓ | ✓ | ✓ | ✓ | 1.080*** | ✓ | ✓ | ✓ | ✓ | ✓ |
| | | | | | | | (0.181) | | | | | |
| **Constant** | 7.854*** | 7.850*** | 7.824*** | 7.851*** | 8.028*** | 7.887*** | 7.852*** | 10.527** | 9.159** | 10.732** | 10.556** | 10.576** |
| | (1.611) | (1.612) | (1.611) | (1.611) | (1.621) | (1.611) | (1.611) | (3.501) | (3.172) | (3.404) | (3.503) | (3.502) |
| Observations | 8,316 | 8,316 | 8,316 | 8,316 | 8,316 | 8,316 | 8,316 | 2,014 | 2,014 | 2,014 | 2,014 | 2,014 |
| N of students | 6,925 | 6,925 | 6,925 | 6,925 | 6,925 | 6,925 | 6,925 | 1,590 | 1,590 | 1,590 | 1,590 | 1,590 |
| Cohen's $d$ effect size of $\beta_1$ | -0.018 | -0.018 | 0.001 | -0.009 | -0.133 | 0.005 | -0.017 | -0.054 | -0.058 | -0.068 | -0.054 | -0.108 |
| The joint linear effect of $\beta_1$ & $\beta_3$ | 0.039 | 0.040 | 0.102 | 0.066 | -0.295 | 0.189 | 0.045 | 0.191 | 0.162 | 0.224 | 0.189 | 0.029 |
| | (0.098) | (0.098) | (0.107) | (0.118) | (0.259) | (0.122) | (0.112) | (0.205) | (0.191) | (0.201) | (0.205) | (0.236) |

All models (Column 1–12) contain the following preregistered standard baseline control variables: student's gender, age, ability, student is a first-year student, the type of training, the financial form of training, the level of training, the difficulty of the exam, and study program fixed effects.

The table lists those variables that we preregistered as a variable to test treatment heterogeneity (Z). Some of the standard control variables are listed in the table as they appear among variables in Z. We marked these variables with the ✓ sign indicating that the given variable was included in the regression even though its estimated coefficient was not included in the table.

In addition to the standard baseline variables, columns 8–12 contain the following preregistered additional baseline variables from the baseline survey, and thus they are available for a subset of students: baseline test anxiety, baseline self-confidence, baseline external control, and parental education. Since all of the additionally used control variables were preregistered as a variable to test treatment heterogeneity (Z), all of them are listed in the table and therefore marked with the ✓ sign.

[a] To enhance readability, the *Interaction* (T×Z) refers to the product of the treatment variable (T) and a specific main effect (Z). The coefficient of the corresponding main effect is shown in the table. For example, in Column 2, the interaction refers to the product of T×Students' ability, and in Column 10, the *Interaction* refers to the product of T× Baseline self-confidence.

[b] z-standardized variable at 0 mean and 1 standard deviation.

Standard errors in parentheses

*** p<0.001

** p<0.01

* p<0.05

+ p<0.1.

**Table 6. Hypothesized treatment heterogeneity.**

| Baseline variables [Z] | *The treatment effect is higher among students* | Primary outcome | Secondary outcomes | | |
|---|---|---|---|---|---|
| | | Grades | Self-efficacy | Motivation | Test anxiety |
| **Test anxiety** | *with high baseline test anxiety* | NS | **Supported** | NS | NS |
| **Self-confidence** | *with low self-confidence* | NS | **Supported** | NS | **Supported** |
| **External control** | *with external control* | NS | NS | NS | NS |
| **Parental education** | *whose parents do not have a university degree* | NS | NS | NS | NS |
| **Students' ability** | *with weaker baseline performance* | **The opposite is supported** | NS | NS | NS |
| **First-year student** | *among first-year students* | NS | NS | **The opposite is supported** | NS |
| **Female** | *among female students* | NS | NS | NS | NS |
| **Has mobile phone** | *receiving a text message on mobile phone* | NS | NS | NS | NS |
| Day of message | *who received the message later* | NS | NS | NS | **Supported** |
| **Exam difficulty** | *who take a difficult exam* | NS | NS | NS | NS |

NS = Not significant.

encouragement message that praised students for their past achievements. Encouragement messages were sent out via two channels: e-mail and SMS text messages.

The field experiment had a crossover design: The treatment and control conditions varied within the same students. A random half of the students received the encouragement message before their first exam and the control message before the second exam. The other half of the students received the same message before their second exam and the control message before the first exam.

Our primary outcome variable was students' end-of-semester exam grades, obtained from the university's registry. We collected secondary outcome variables via an endline survey that both the treated and control students voluntarily answered before their exam. The subsample

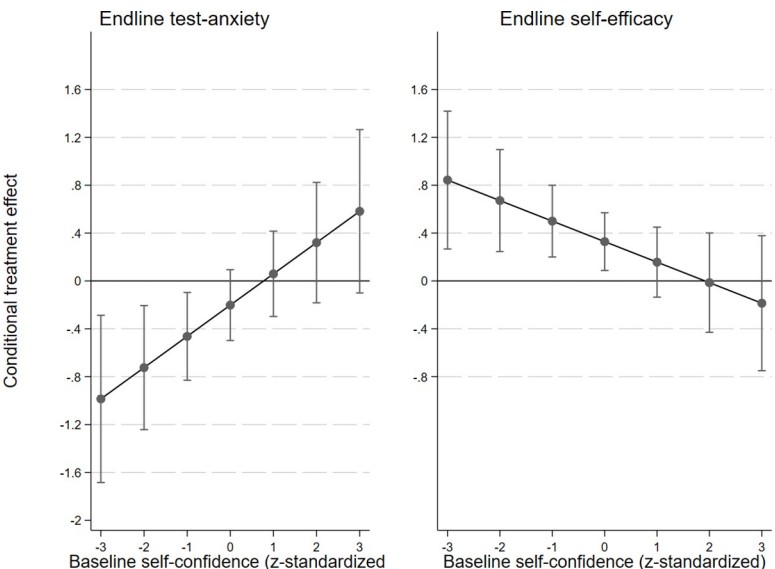

**Fig 7. The explored conditional heterogeneous treatment effect (y-axis) on students' endline test anxiety (left panel) and endline self-efficacy (right panel), based on students' baseline self-confidence (x-axis).** The left panel: corresponds to Model 10 in Table 5, N of observations = 2,014; N of respondents = 1,590. The right panel: corresponds to Model 10 in Table 6, N of observations = 2,016; N of respondents = 1,594.

of students who answered the endline survey consisted of a more advantaged group of students regarding their baseline data, e.g., in terms of students' ability. Since we found little treatment heterogeneity in the secondary outcomes according to students' observed baseline variables, the potential main treatment effect in the whole analytic sample may have a similar size to the effects we observed among the more advantaged subsample of those who answered the endline survey.

Overall, our analysis provides new answers in several aspects. First, we revealed that encouraging students shortly before their exams with automated messages praising past achievements influenced students' self-efficacy but had no or limited effect on their test anxiety and motivation. Therefore, our results suggest that only self-efficacy is malleable and can be impacted by positive feedbacks [48, 49]. However, the development of students' test anxiety or motivation requires a different treatment.

Second, encouraging students with automated messages shortly before exams does not affect exam grades. Therefore, experimentally induced self-efficacy does not translate to higher exam grades. We precisely estimated a treatment effect close to zero with small standard errors. Thus, our results conflict with prior findings that conclude that encouragement increases students' test performance [30, 31]. The nil treatment effect on students' grades could be impacted by the grading-on-a-curve effect, which impedes the observation of the treatment effect of any intervention targeting students' grades. Specifically, if the distribution of the grades is fixed (e.g., teachers distribute a fixed and constant number of each grade), then any increase in students' grades from the treatment would not be revealed.

Third, scaling up similar encouragement campaigns might have limitations since it only impacts more able students' exam grades. Thus, the success of prior interventions with a similar scope among a specific group of students cannot be generalized to the average university student [30, 31].

Our findings have two important implications that warrant further consideration. First, encouraging words boost students' self-efficacy. Before exams, students receive different "messages" from their teachers, parents, and peers, concerning their ability, performance, and chances of success. Depending on the tone of these messages, each of them might increase or decrease students' self-efficacy. Our experiment reveals that students are sensitive to these words. Therefore, teachers, parents, and peers should be careful with their statements and words since these words are not *just* words but also affect students' self-efficacy.

Second, academic performance among students with initially low ability cannot be raised merely by encouragement. The encouragement instead provides a small lift in more able students' exam grades.

There are several possible explanations why we found that the intervention only affected the exam grades of more able students. More able students might be more motivated [50]. By contrast, less able students may be less interested in gaining a good grade at the exam, and therefore not sensitive to the treatment.

Another possible reason is that students with lower baseline abilities may have less confidence in their abilities [51]. Therefore, they might not believe that the encouragement message is addressed to them. In particular, students with lower ability may achieve lower grades at university. They could falsely conclude that they are not successful and regard the message that praised past achievement as not relevant. By contrast, more able students who achieve better grades might subjectively rate themselves as more successful and therefore place greater trust in the encouragement message.

Finally, the encouragement message might help students to recall better the knowledge they have already obtained. Since students received the message shortly before the exam, it could not have increased their effort to acquire more knowledge, but the message could fine-tune

how students access their existing knowledge. More able students might be better prepared for the exam and have more knowledge to mobilize when they receive encouragement. By contrast, students with initially lower ability may be less prepared and have less knowledge to recall. Therefore, the existing difference in students' knowledge might explain how much benefit they could gain from the encouragement.

Overall, we interpret our results on the main treatment effects within the framework proposed by Jacob et al. [52] of learning from null results in three respects. First, one should consider the typical potential growth in students' exam grades over the intervention period. In our case, the intervention period is a couple of hours (i.e., the time elapsed between the time students received the message and the exam). Within such a short period, one should not expect large changes in students' knowledge (that could be translated into higher grades). Therefore, the impact of any intervention (and not just particularly our encouragement campaign) that targets students a couple of hours before their exam might have a limited effect on students' outcomes. Thus, the precisely estimated zero results in exam grades, which suggests that the intervention had no practical significance for students' exam grades, could be attributed to the short period of time and (in addition) the light-touch (nonintensive) intervention.

Second, one should consider the theory behind the outcomes. In our case, any change in students' exam grades can be solely attributed to the change in a student's ability belief due to the encouragement message. By contrast, changes in the secondary outcomes can be attributed to the encouraging words that students received in the treatment message. Therefore, our results indicate that experimentally induced ability beliefs (such as self-efficacy, motivation, and test anxiety) do not translate to higher cognitive performance in the short run. Nevertheless, encouraging words do affect some ability beliefs, such as self-efficacy beliefs.

Lastly, one should consider the cost of the treatment. A low-cost intervention with a small impact might be considered successful despite the size of its impact, specifically due to the low costs. We invested about 210 USD (60,000 HUF) in sending out the text messages; sending out the e-mails had no incidental costs. For this level of investment, a short-lived gain in students' self-efficacy is a substantial achievement, even though it does not directly boost students' exam grades.

Nevertheless, instead of encouragement, policymakers and educational planners should investigate other means to motivate low-ability students, as their exam grades seem to be resistant to encouragement. Providing useful information for organizational and time management [53] and gamification [54] are techniques that have been successfully used to target students with initially low ability in prior practice and research.

Future encouragement interventions should further improve on our automated encouragement message, which required minimal additional human effort from the message provider. For example, personalized (rather than uniform) messages sent by senders to whom students have contact (e.g., a professor or role model rather than the Head of the Directorate of Education, with whom most students do not have direct contact) could increase the efficacy of future treatments. Furthermore, interventions that encourage students earlier, or more consistently throughout the semester on a systematic rather than occasional basis, should also be considered to increase the treatment effect.

In sum, we conclude that automated encouragement messages sent shortly before students' exams are not a panacea for increasing students' academic achievement. However, students' self-efficacy is sensitive to encouraging words, even if these words arrive shortly before an academically challenging exam situation. Therefore, further encouragement interventions targeting students' self-efficacy might promote a school climate that boosts students' engagement in the academic side of school life [3, 4].

## Supporting information

**S1 Appendix. Students' perceptions of the intervention.**
(DOCX)

**S2 Appendix. Subsamples.**
(DOCX)

**S3 Appendix. Descriptive statistics of the outcome variables in the whole sample and in the subsample of those who answered the endline questionnaire.**
(DOCX)

**S4 Appendix. Control variables.**
(DOCX)

**S5 Appendix. Pairwise correlation between various psychological measures and the secondary outcome variables.**
(DOCX)

**S6 Appendix. Results of the alternative model specifications.**
(DOCX)

**S7 Appendix. Results of sensitivity analyses.**
(DOCX)

**S8 Appendix. The original Hungarian version of various survey instruments.**
(DOCX)

**S9 Appendix. The English version of various survey instruments.**
(DOCX)

**S10 Appendix. Deviations from the preregistered pre-analysis plan.**
(DOCX)

## Author Contributions

**Conceptualization:** Tamás Keller, Péter Szakál.

**Data curation:** Tamás Keller.

**Formal analysis:** Tamás Keller.

**Funding acquisition:** Tamás Keller.

**Investigation:** Tamás Keller.

**Methodology:** Tamás Keller.

**Project administration:** Péter Szakál.

**Software:** Tamás Keller.

**Visualization:** Tamás Keller.

**Writing – original draft:** Tamás Keller.

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
