## [Decision Letter · Decision Letter 0]

15 Mar 2021

PONE-D-21-03213

Not just words! Effects of encouragement on students’ exam grades and non-cognitive skills—lessons from a large-scale randomized field experiment

PLOS ONE

Dear Dr. Tamás,

Thank you for submitting your manuscript to PLOS ONE. After careful consideration, we feel that it has merit but does not fully meet PLOS ONE’s publication criteria as it currently stands. Therefore, we invite you to submit a revised version of the manuscript that addresses the points raised during the review process.

The paper is very interesting and in general well executed. However, as the three excellent reports point out, there are many important issues that need to be clarified. The authors must consider carefully all the points suggested by the referees. Importantly, there are some interpretations of the coefficients that can be misleading, some results that seem not so clear as reported (regarding student intentions to do well), and several aspects that need to be better justified. The authors must clarify the different concerns raised by the referees and consider and discuss their suggestions, that I also believe are helpful to understand better the different aspects of the paper. I know this will require a huge effort, so you consider that need additional time please do not hesitate to ask for it. 

We look forward to receiving your revised manuscript.

Kind regards,

Alfonso Rosa Garcia

Academic Editor

PLOS ONE

Journal Requirements:

2. Please include additional information regarding the survey or questionnaire used in the study and ensure that you have provided sufficient details that others could replicate the analyses.

For instance, if you developed a questionnaire as part of this study and it is not under a copyright more restrictive than CC-BY, please include a copy, in both the original language and English, as Supporting Information.

Reviewers' comments:

Reviewer's Responses to Questions

**Comments to the Author**

1. Is the manuscript technically sound, and do the data support the conclusions?

Reviewer #1: Partly

Reviewer #2: Partly

Reviewer #3: Yes

2. Has the statistical analysis been performed appropriately and rigorously? 

Reviewer #1: Yes

Reviewer #2: I Don't Know

Reviewer #3: Yes

3. Have the authors made all data underlying the findings in their manuscript fully available?

Reviewer #1: Yes

Reviewer #2: No

Reviewer #3: Yes

4. Is the manuscript presented in an intelligible fashion and written in standard English?

Reviewer #1: Yes

Reviewer #2: Yes

Reviewer #3: Yes

5. Review Comments to the Author

Reviewer #1: Please see the attached document called "referee report".

Reviewer #2: Major comments: 1 The message sent to students includes promise of a possible lottery game prize. Do you think this had any effect on students behavior towards the message?

2 The follow-up qualitative survey appears to have great attrition. Is it possible this had an effect on findings? Was there differential attrition by treatment and control samples?

3 When describing your treatment variable in the first paragraph of the 7th page it is very confusing. Particularly "regardless of whether they had received it before the first or before the second exam.". Do you mean that treatment is 1 for the first exam for A and 0 for B, and then 1 for the second exam for everyone? Or do you mean something different? Please clarify.

4 Why do you deploy multi-level random effects models here? I am unfamiliar with the technique for randomized control trials? You have a randomized sample. One table should be just the difference between treatment and control since randomization ensures treatment and control samples are balanced and one group is not treated on the first test (group B). No need to employ complicated models with randomization on the first test.

5. You use an interaction with treatment and the second test in EQ1. Are the treatment effects reported the effects on the first test then? Where can I find the interaction results for the treatment effect and the second test?

6. The results in column (2) of table 4 are confusing to me. Why does the coefficient on the treated variable barely change between column (1) and (2) given that the interaction is positive and significant? Shouldn't the treatment effect be pulled down by the positive interaction since it now represents the treatment effect for low ability students? You say in the footnotes to the table that all models include controls for ability but then show the coefficient for ability in column (2) but not (1). Does this have something to do with multi-level random effects models? What am I missing here?

Minor comments: 1 Full paragraph 2 page 4, last sentence needs clarification. 2. Last full paragraph, page 5, massages should be messages.

Reviewer #3: Review PONE-D-21-03213

Not just words! Effects of encouragement on students’ exam grades and non-cognitive skills – lessons from a large-scale randomized field experiment

General comments

This study shows results of a large-scale randomized field experiment targeted at students’ exam grades, as well as their test anxiety, self-confidence and intention to do well on a test, by using automated encouragement messages. The experiment was pre-registered. No average treatment effects were observed on exam grades, yet the intervention did show some effects on the non-cognitive skills (i.e. self-confidence and intention to do well). There also seems some heterogeneity according to the ability level of the students.

The study touches upon an important aspect of learning behavior and academic performance. Not only is the importance of well-developed non-cognitive skills (among which self-concept, ability to deal with anxiety and aspirations, which are targeted in this study), next to cognitive skills, for academic success and life outcomes well-documented, there is also a growth of so-called social-emotional learning programs that address the development of such skills in school. It is important to understand what works well and which incentives have no effects. The use of large-scale RCT’s in the field are of great importance to this. Yet these are not easy to develop, and the authors took the courage to undertake such a large-scale field experiment. The intervention, in turn, is an easy to implement one in educational practice if proven effective.

The study is well performed with a rigorous design and methodological approach, and the article is generally well-written. However, I think it can be sharpened a bit before publication. For example, I think the study can be somewhat bit stronger embedded in the literature. There is quite some research on the effectiveness of social-emotional learning programs or on the role of confidence nudging in relation to performance, that relates to this I guess. See some further comments below. I also recommend that the authors take a close look at how the information about the sample, procedures and measures is given, I sometimes had a difficult time grasping the details and keep the focus.

Comments per section

Introduction:

• p2, par3: The authors might want to take a look at a paper of Tenney et al. (2015) who investigate the relation between optimism and performance with a range of (small-scale) experiments, including some of which try to impact people’s optimism by encouragement/discouragement messages. For example, they observe that manipulated optimism affected their participants’ self-reports of felt optimism and a behavioral measure of their persistence, which are in turn important for performance.

• p2, par4: The authors mention that there is significant heterogeneity in the effect sizes observed in previous studies. I feel that they might elaborate a bit more these differences than is so far done in this paragraph. Now only the size and more general framework that the studies take are mentioned, but are there also some conclusions with respect to subgroups for example?

• p2, par5: I am a bit puzzled by this argument. I understand that it might slow down the observed effects in studies, but in the end we want effective strategies to be integrated in teacher practice, right? I think you mean that the interventions proposed in the literature require more of an overhaul of the system. And that this might not always be feasible or desirable. But that there is a lack of studies on more easy to apply measures that could be integrated in education, independent of teacher motivation or experience. Perhaps I am misreading this, but the authors might want to explain a bit.

• p2, par6: In my view there might be an additional concern prevalent when looking at the current literature and that is the lack of large-scale field RCT’s. Many of the experimental studies in these fields are either in a lab setting, or using small samples in the field if I am not mistaken. Studies in the field are mostly non- or quasi-experimental to my knowledge. The authors can correct me if I am wrong. Perhaps this might be added as an additional concern, also showing the contribution of this current paper.

• p3, par6: there is some discussion going on in the (education) literature on the effect sizes (small or null) in field experiments. The last two paragraphs of the discussion of a recent paper by Feron & Schils (2020) touch upon this issue and you might find this interesting for your study.

Design, data and method

• p4, par7: very minor query, but what kind of things can be bought in the SZTE gift shop? This might give some information about the incentive and to what extent it is a real incentive/reward.

• p5, par6: Do you know how many students know that they did not receive the encouragement message? From those only 17% was sad/very sad, right? Is it the 33% mentioned in the next paragraph? This gives a bit more insight in the extent to which we can agree that that likelihood of adverse treatment effects is ‘moderate’, as you state.

• P7, par1: Perhaps you can already mention here that the first and second exams are in different subjects, because when I was reading this paragraph it was unclear to me why you did not distinguish between whether they received the message for the first or for the second exam? The information about the differences between the first and second exams, as well as information on the general exam system in Hungary follows later, but the reader might already be a bit puzzled. It is many details to digest.

• p8, par 1: how much time is there between the pre- and post-test? Is it a reasonable period to expect effects?

• p8, par 3, you might not know, but might the missings due to illness be related to test anxiety? If you have any information on this, that would be useful, e.g. perhaps those that scored high on test anxiety in the survey are more often absent?

• p8, footnote: ether > either.

• p9, point 3: I am bit surprised by the locus of control, this is not mentioned in the literature. Perhaps the authors can address it in the literature, so the reader understands why it is included.

Results

• p13, par 6: hypostatized > hypothesized.

Discussion

• p15, par 5 and p16, par 6: I was just wondering about the effect of the treatment on exam grades, these are only given in 1 2 3 4 5, right? In that case the treatment should be really strong to see an effect on grades? Or am I misinterpreting the grading system? It might be that in the previous literature the grading system used was different and allowed for ‘easier to establish’ effects?

• p15, par 6 and later when you discuss this more thorougly in the discussion: this result for the high able students might indeed link up to boosting confidence that increases the grades. (it relates to the general effect on self-confidence, you observe). They already knew they were good (or among the upper part of the ability distribution) and receiving an encouragement message basically confirms that feeling and they even get more confident in that they will succeed in the exam. Perhaps the psychological literature on (over)confidence might be useful here, you might want to check out papers of Don Moore, who wrote about this. The low ability students might indeed have given up, and have become rather ignorant to studying and performing well on tests. While it is quite important also for their future training participation as many studies show that low-educated/ability workers are less prone to investing in further training during the life course. More emphasis might be put on understanding the mechanisms behind the non-effects of encouragement among low-ability students. However, having said that, I think the conclusions on the heterogeneity by ability should be modest, as the observed effects were only marginally significant. Moreover, we are talking about low-ability students in a university setting, so not overall low-ability students, i.e. those that already made it to an academic study. I think this is important to mention.

• p16, par 6: perhaps the effect on the non-cognitive skills needs more time to translate to cognitive skills, have you considered that? I would at least say it did not translate into short-run cognitive results.

I hope the authors can use my comments and suggestions to further improve the paper.

References mentioned in this review:

Tenney, E. R., Logg, J. M., & Moore, D. A. (2015). (Too) optimistic about optimism: The

belief that optimism improves performance. Journal of Personality and Social Psychology.

Feron, E. & Schils, T. (2020) A randomized field experiment using self-reflection on school behavior to help students in secondary school reach their performance potential”, with Eva Feron. Frontiers in Psychology – Personality and Social Psychology.

6. PLOS authors have the option to publish the peer review history of their article (what does this mean?). If published, this will include your full peer review and any attached files.

Reviewer #1: No

Reviewer #2: **Yes: **Daniel Dench

Reviewer #3: **Yes: **Prof. dr. Trudie Schils

---

## [Author Response · Author response to Decision Letter 0]

25 May 2021

See also as Response to Reviewers among the uploaded ducuments

Reviewer #1

A. Paper Summary

This paper experimentally evaluates the effects of sending encouragement messages to students via SMS and e-mail on exam performance, test anxiety, self-confidence, and intention to do well on the exam. The authors conduct the experiment at the University of Szeged with approximately 15,000 students. The encouragement messages are sent to (a randomly selected) half of the students before the first exam and to the other half of the students before the second exam. Students are also asked to complete a three-question survey, asking about anxiety, self-confidence, and intentions, prior to taking both exams. The authors find no effect of receiving the encouragement message on exam performance or test anxiety. They do find that treatment increased students’ self-reported self-confidence and intention to do well before the exam. 

B. Evaluation 

This is a well-motivated, ambitious paper that explores the important question of whether we can improve students’ noncognitive skills cheaply and at scale. The data gathering effort is impressive and the analysis is thoughtful. Indeed, the paper offers many interesting results to contemplate and there is a lot to like about the paper. 

I do, however, have several comments and suggestions. 

Major Comments: 

1. The Messages Students Received. 

a. General Content Selection. It would be good to offer more discussion on how the language in the e-mail and text messages was chosen. These messages aim to affect student exam performance by first affecting self-confidence, test anxiety, and intentions to do well. How does the language in the messages affect each of these three traits? 

The first sentence of the e-mail message praises students for their prior achievements (“you already have many successful exams behind you”). The sentence confirms students’ competence, and empowers them by pointing to their successes rather than their challenges. This sentence, therefore, is intended to raise students’ self-efficacy as, according to Bandura, (1977), accomplishments of past performance and verbal persuasions are important sources of self-efficacy. The sentence also aims to influence students’ test anxiety since positive affirmation messages decrease students’ worries (Deloatch et al., 2017). The sentence is valid for all students, since students have already taken successful exams to be admitted to the university. 

The second sentence signals trust in students’ success (“I truly hope that you will succeed”). The sentence is designed to be a self-fulfilling prophecy (Rosenthal & Jacobson, 1968). It is intended to affect students’ behavioral intention (Ajzen, 1991) by evoking their motivation to fulfill the meaning of the sentence (Friedrich et al., 2015; Rosenthal & Jacobson, 1968).

The SMS messages contain the same elements (praise for past achievements and trust) in a more condensed form

b. SMS Content and Length. Why not include more motivating content into a slightly longer text message and/or include the link to the three-question survey in the text message instead of just the e-mail? My experience is that students are much more likely to engage with text messages than e-mails. 

The majority of students (66%) received the treatment SMS 3 hours before the exam Since students are unlikely to answer a questionnaire just before their exam, therefore, only the e-mail contained the link to the online questionnaire. 

SMS text messages were shorter than the e-mail messages since the SMS package the university ordered made it available to send text messages with a fixed length, e.g., the numbers of characters were given. 

c. Timing of Messages. Could the authors offer more discussion on when (exactly) students received the messages? Figure 2 is instructive, but I am not sure how to read it. Does it mean that exams could have happened on Dec 9, 2019, Dec 19, 2019, etc.? If so, then there are six exam dates and that would mean that some students received their message on the day of the exam while others received it up to nine days before the exam or the day after the exam. I would like to know when, relative to the day of the exam, students received the encouragement messages. (Figure 4 implies that the average student completed the three-question survey 13 hours before the first exam. But, again, when were these messages sent to students?) That seems important for thinking about the treatment effect. 

Motivated by your excellent suggestion, we have elaborated more on Figure 2, which is now Figure 3. The Figure shows the total number of treatment messages (e-mail and SMS) corresponding to an exam on a particular calendar date. The x-axis lists all the exam days within the exam period. Approximately 80% of treatment messages were sent out in the first ten days of the campaign. This indicates a condensed treatment period, mainly concentrated in the first few days of the exam period.

We do not know exactly when students read the treatment messages—that is, how long before the exam. Nevertheless, the date when students filled in the endline survey indicates when they might have read the e-mail. Figure 1 shows when students completed the endline survey relative to the corresponding exam. On average, students filled in the questionnaire 13 hours before their exam. This means that the treatment e-mail targeted the students a couple of hours before their exam. 

Figure 2 shows the time (in hours) relative to the exam when the treatment SMS was sent out to students’ mobile devices. The majority of students (66%) received the treatment SMS 3 hours before the exam, indicating that we encouraged students shortly before their exams. 

2. Estimation and Interpretation of Treatment Effects. 

a. Main Estimating Equation. Could the authors spend more time discussing how to interpret the parameters 𝛽1, 𝛽2, and 𝛽3 in equation 1? I do not believe the interpretation is as easy as the authors make it out to be, given the crossover randomization design where all students are sometimes treated and sometimes control. Here is a basic difference-in-differences style table that one would normally use when time is interacted with treatment status (ignoring other covariates in the model) but considering the crossover design in the current experiment: 

A couple of challenging questions of interpretation come up: 

• 𝛽2: This is the difference between the average second exam score of group A and the average first exam score of group B. As such, this will only capture the (pure) difference in difficultly between the exams when the persistence of the treatment effect is exactly zero. Otherwise, some of the treatment effect from the first exam will persist into the second exam, affecting the performance of group A on the second exam. At the extreme, suppose treatment perfectly persists to the second exam and the exams are exactly the same level of difficulty. Then we should have 𝛽1= 𝛽2. Given this, is it not 𝛽2 that captures carry-over effects instead of 𝛽3? Perhaps I misunderstood what is meant by “carry-over”. 

• 𝛽3: This parameter is difficult to interpret. Isolating it requires summing up the average first and second exam scores for Group B and Group A and then taking the difference between these sums: 

So, then, how do we interpret this parameter? 

Thank you for raising these excellent points that motivated us to elaborate more on the interpretation of the coefficients. 

The coefficients in Eq.1 are unstandardized regression coefficients. The coefficient β_1 identifies the causal treatment effect. The coefficient is the mean difference in the first exam grades between students in the treated minus the control condition. 

The coefficient β_2 identifies the period effect, i.e., the difference in exam grades between the first and second exams. The coefficient does not have a causal interpretation, since the ordering of students’ exams was not randomized. The coefficient is the mean difference in control students’ exam grades (the difference in mean grades control students earned at the second minus the first exam). 

The coefficient β_3 identifies the carry-over effect, i.e., the difference in exam grades between the students in the treated and control conditions in the first and second exams. The coefficient is the difference of two mean-differences, i.e., the mean difference of exam grades between treated and control students in the second exam minus the mean difference of exam grades between treated and control students in the first exam. 

If there is a statistically significant β_3coefficient, students’ treatment before their first exam has a long-lasting effect or long wash-out period. In other words, a significant carry-over effect reflects that encouraging students before their first exam affects their grades at the second exam; thus, the ordering of the treatment matters. A significant carry-over effect biases the estimation of the average treatment effect (Piantadosi, 2005). 

Our hypothesis on the main treatment effect will be confirmed if we obtain a positive coefficient for β_1, and if we do not have a carry-over effect—i.e., if the main treatment effect concerning students’ first and second exams do not differ statistically. 

It may be helpful to report estimates from two separate regressions: one where the first exam is the dependent variable; and another where the second exam is the dependent variable. This would not solve the issues of treatment dynamics and persistence, but it might make it easier for the reader to think about the estimated parameters. 

• A general comment on the treatment effects estimation: Ding and Lehrer (2010) is an excellent piece on estimating dynamic treatment effects and interactions between treatment effects in multiple time periods. Doing this fully requires four groups of students: (i) never treated, (ii) treated in period 1 but not 2, (iii) treated in period 2 but not 1, (iv) treated in both periods. Here, the authors do not have a never-treated group or an always-treated group, making it impossible to recover some of the effects I think they would like to recover. 

I may be wrong about how the authors are interpreting the parameters and what is meant by “carry-over” effect. In that case, it would be good for them to clarify. 

Thank you very much for suggesting us Ding and Lehrer’ excellent paper. As a robustness check, we have estimated the main treatment effect with exam-subject fixed separately on students’ first exam (Table A1 in Appendix F) and second exam (Table A2 in Appendix F). We have not included these tables in the main text since they are based on different models and not on those that we preregistered. Table A5-A8 in Appendix G shows the results on “always-treated”, e.g., on those students who have two outcomes: concerning the first and second exams. The results of these tables are qualitatively similar to those results we show in the main body of the manuscript. 

b. selection into endline questionnaire. Table 3 is important and informative, showing no differential selection between groups A and B, except into completing the endline survey twice versus once. But did the authors check whether students were more likely to complete the endline questionnaire before the exam on which they were treated? That is, was group A more likely to complete the endline survey before exam 1 than group B? And was group B more likely to complete the survey before exam 2 than group A? 

I do not believe any of the results in Table 3 speak to this (Appendix Tables A7 to A10 do somewhat), although I could be wrong. This type of selection is important to rule out because it would imply students differentially selecting into the survey based on treatment status, calling into question the estimated treatment effects on the items from the survey. 

The treatment status significantly decreased students’ willingness to answer the endline questionnaire, both before students’ first and second exams by 3.6 and 5.2 percentage points, respectively. As the e-mail that the control students received prompted them to go directly to the lottery, control students received stronger incentives to participate in the survey and win, which might explain why control students were more likely to fill in the endline survey. This type of selection could undermine the results on the secondary outcomes. Nevertheless, as Tables A6 to A8 Appendix G show, the estimations were qualitatively similar among those who answered the endline questionnaire twice, and thus filled in the questionnaire in the treated and also in the control condition. 

c. Interpreting Treatment Effects Considering Possible Contamination. The authors acknowledge the possibility of treatment contamination on pages 5 to 6 and present compelling evidence from the online survey to suggest that contamination likely does not affect the estimates much. But I am wondering why this survey was done five months after the encouragement campaign. Can the authors argue that students are still likely to remember how they felt when they were not receiving messages and their friends were? 

Ideally, the online follow-up survey should be administered earlier, immediately after the treatment. It was not feasible, however, due to the closures and switch to online education caused by the COVID-19 pandemic. These changes challenged the university’s online platform and required the full attention of the administrative staff who could administer the infrastructure of such an online survey. 

Although five months is a significant amount of time and students’ memories might be attenuated, 79% of the respondents correctly recalled the content of the message, while 9.5% of students claimed not to remember. The rest of the respondents either did not answer the question (6.5%) or recalled incorrect content (5%). These figures indicate that students’ memories about the intervention had not attenuated significantly by the time of the follow-up survey. 

d. Subgroup Effects. There are many regressions run. Between Tables 4 to 7, there are 48 columns. Table 8 lists 40 hypotheses to be tested. On page 15, the authors acknowledge that the sporadic significant interaction coefficients across these tables could simply arise from multiple testing. I think it would be good to include such cautious language when discussing the interaction coefficients in the body of the paper, too. Otherwise, I think these statements sound too conclusive. 

Thank you very much. We have toned down the langue about the interaction in the discussion. We explicitly state in the text that the results are exploratory due to multiple testing.

e. Reported Effects on Student Intentions to Do Well. Table 7 shows the effect on intentions did not replicate in the second exam because 𝛽1+𝛽3 (the difference in average exam 2 scores between group B and group A) is not statistically different from zero. One might read this as a failed replication attempt, depending on how we should think about treatment persistence and time, and so I am not sure why the effect on students’ intentions to do well is treated as a headline result (mentioned in the abstract and introduction) when the finding may not replicate within the paper. 

Thank you very much. We have toned down the langue when discussing the results and deleted the reference from the abstract and introduction. Both in the abstract and in the introduction, we write that in the case of students’ motivation, the treatment effect is most evident in students’ first exam but is attenuated in their second exam. Thus the treatment effect was not replicated in the second exam. 

3. Cost-Benefit Conclusions. 

On page 16, the authors note that their campaign produced cost-effective results on secondary outcomes. Could they provide more evidence for the benefits side of this claim? There are two main concerns. First, as mentioned above, it is questionable whether treatment had any effect on intentions to do well, as this result does not seem to replicate on exam 2. So that only leaves the self-confidence result as robust. Second, then, what is the benefit of the (likely) short-term boost in self-confidence? How does one quantify it and why is it valuable? 

Your excellent point has motivated us to streamline our argument. In short, encouraging students systematically and not just shortly before their exams is a possible school practice that can forge positive emotional involvement and engagement with the academic aspect of school life. Therefore, light-touch encouragement interventions might have substantial significance in themselves, even though these interventions do not directly affect students’ exam grades.

We interpret our results on the main treatment effects within the framework proposed by Jacob et al. (2019) of learning from null results. First, one should consider the typical potential growth in students’ exam grades over the intervention period. In our case, the intervention period is a couple of hours (i.e., the time elapsed between the time students received the message and the exam). Within such a short period, one should not expect large changes in students’ knowledge (that could be translated into higher grades). Therefore, the impact of any intervention (and not just particularly our encouragement campaign) that targets students a couple of hours before their exam might have a limited effect on students’ exam grades. Thus, the precisely estimated zero results in exam grades, which suggests that the intervention had no practical significance for students’ exam grades, could be attributed to the short period of time and (in addition) the light-touch (nonintensive) intervention. 

Second, one should consider the theory behind the outcomes. In our case, any change in students’ exam grades can be solely attributed to the change in a student’s ability belief targeted by the encouragement message. By contrast, changes in the secondary outcomes can be attributed to the encouraging words that students received in the treatment message. Therefore, our results indicate that the positive beliefs we experimentally induced by the encouragement intervention do not translate into higher cognitive performance in the short run. Nevertheless, encouraging words do affect self-efficacy. 

Lastly, one should consider the cost of the treatment. A low-cost intervention with a small impact might be considered successful despite the size of its impact, specifically due to the low costs. We invested about 210 USD in sending out the text messages; sending out the e-mails had no incidental costs. For this level of investment, a short-lived gain in students’ self-efficacy is a substantial achievement. Further, the implementation of the intervention does not require additional human effort; it could be scaled up to a virtually unlimited number of students. These features suggest that similar interventions can be worthwhile despite not directly boosting students’ exam grades. 

In sum, we conclude that automated encouragement messages shortly before students’ exams are not a panacea for increasing students’ academic achievement. However, students’ self-efficacy is sensitive to encouraging words, even if these words arrive shortly before an academically challenging exam situation. Thus, sending out encouraging messages shortly before students’ exams on a systematic rather than occasional basis might be a cost-effective tool for boosting students’ self-efficacy. Therefore, encouragement interventions might help to create a school climate that boosts students’ self-determination in the academic side of school life. They may thus have their own substantive importance (Appleton et al., 2008; Christenson et al., 2012). 

Minor Comments: 

1. Last sentence of the abstract: “The sporadic treatment effect heterogeneity in the secondary outcomes…” The word “secondary” should actually be “primary” I believe. 

Thank you, we have changed the abstract

2. At the end of page 2, the authors say that the study is not specific to a particular sub-population of students. That is true but it is specific to the types of students who attend a given university. I would probably just be more conservative with this statement, as the study does not stretch across many institutions. 

Thank you for your suggestion. We have toned down the language.

3. Page 4, the last sentence of section II.2: in which units is the 0.03 effect size measured when discussing power calculations? 

We have changed the text and clarified that the corresponding effect size is Cohen’s d effect size effect size

 

Reviewer #2: 

The message sent to students includes promise of a possible lottery game prize. Do you think this had any effect on students behavior towards the message?

The treatment status significantly decreased students’ willingness to answer the endline questionnaire, both before students’ first and second exams by 3.6 and 5.2 percentage points, respectively. As the e-mail that the control students received prompted them to go directly to the lottery, control students received stronger incentives to participate in the survey and win, which might explain why control students were more likely to fill in the endline survey. This type of selection could undermine the results on the secondary outcomes. Nevertheless, as Tables A6 to A8 Appendix G show, the estimations were qualitatively similar among those who answered the endline questionnaire twice, and thus filled in the questionnaire in the treated and also in the control condition. 

The follow-up qualitative survey appears to have great attrition. Is it possible this had an effect on findings? Was there differential attrition by treatment and control samples?

Participation in the follow-up survey was voluntary. Since the follow-up survey provides qualitative insight into students’ perception of the survey, the numbers of respondents do not influence the robustness of the results. Motivated by your valuable comment, however, we have moved the section about the follow-up survey into Appendix A, which decision greatly increased the paper’s readability. 

When describing your treatment variable in the first paragraph of the 7th page it is very confusing. Particularly “regardless of whether they had received it before the first or before the second exam.”. Do you mean that treatment is 1 for the first exam for A and 0 for B, and then 1 for the second exam for everyone? Or do you mean something different? Please clarify.

Thank you very much for your suggestion. We have clarified the corresponding sentence: The treatment variable (T) is a 0/1 variable that indicates whether the student received the encouragement message (T=1), i.e., an e-mail and SMS before the exam. The treatment variable is coded as zero (T=0) if students received the control message, which is an e-mail without encouragement, before their exam. 

Why do you deploy multi-level random effects models here? I am unfamiliar with the technique for randomized control trials? You have a randomized sample. One table should be just the difference between treatment and control since randomization ensures treatment and control samples are balanced and one group is not treated on the first test (group B). No need to employ complicated models with randomization on the first test.

Thank you for this comment. Table 1 shows the balance between those randomized into Group A and Group B. 

You use an interaction with treatment and the second test in EQ1. Are the treatment effects reported the effects on the first test then? Where can I find the interaction results for the treatment effect and the second test?

Thank you for raising these excellent points that motivated us to elaborate more on the interpretation of the coefficients. 

The coefficients in Eq.1 are unstandardized regression coefficients. The coefficient β_1 identifies the causal treatment effect. The coefficient is the mean difference in the first exam grades between students in the treated minus the control condition. 

The coefficient β_2 identifies the period effect, i.e., the difference in exam grades between the first and second exams. The coefficient does not have a causal interpretation, since the ordering of students’ exams was not randomized. The coefficient is the mean difference in control students’ exam grades (the difference in mean grades control students earned at the second minus the first exam). 

The coefficient β_3 identifies the carry-over effect, i.e., the difference in exam grades between the students in the treated and control conditions in the first and second exams. The coefficient is the difference of two mean-differences, i.e., the mean difference of exam grades between treated and control students in the second exam minus the mean difference of exam grades between treated and control students in the first exam. 

If there is a statistically significant β_3coefficient, students’ treatment before their first exam has a long-lasting effect or long wash-out period. In other words, a significant carry-over effect reflects that encouraging students before their first exam affects their grades at the second exam; thus, the ordering of the treatment matters. A significant carry-over effect biases the estimation of the average treatment effect (Piantadosi, 2005). 

Our hypothesis on the main treatment effect will be confirmed if we obtain a positive coefficient for β_1, and if we do not have a carry-over effect—i.e., if the main treatment effect concerning students’ first and second exams do not differ statistically. 

The results in column (2) of table 4 are confusing to me. Why does the coefficient on the treated variable barely change between column (1) and (2) given that the interaction is positive and significant? Shouldn’t the treatment effect be pulled down by the positive interaction since it now represents the treatment effect for low ability students? You say in the footnotes to the table that all models include controls for ability but then show the coefficient for ability in column (2) but not (1). Does this have something to do with multi-level random effects models? What am I missing here?

Thank you very much for your clarification. You are right, all models included the same set of control variables, and thus the tables were misleading. We have corrected this mistake. As a note, below each table, we deployed the following text: All models (Column 1-12) contain the following preregistered standard baseline control variables: student’s gender, age, ability, student is a first-year student, the type of training, the financial form of training, the level of training, the difficulty of the exam, and study program fixed effects. Some of these standardly used control variables are listed in the table: these are marked with the ✓ sign. 

Minor comments: 1 Full paragraph 2 page 4, last sentence needs clarification. 2. Last full paragraph, page 5, massages should be messages.

Thank you very much for your careful reading; we have corrected these mistakes. 

 

Reviewer #3: 

General comments

This study shows results of a large-scale randomized field experiment targeted at students’ exam grades, as well as their test anxiety, self-confidence and intention to do well on a test, by using automated encouragement messages. The experiment was preregistered. No average treatment effects were observed on exam grades, yet the intervention did show some effects on the noncognitive skills (i.e. self-confidence and intention to do well). There also seems some heterogeneity according to the ability level of the students.

The study touches upon an important aspect of learning behavior and academic performance. Not only is the importance of well-developed noncognitive skills (among which self-concept, ability to deal with anxiety and aspirations, which are targeted in this study), next to cognitive skills, for academic success and life outcomes well-documented, there is also a growth of so-called social-emotional learning programs that address the development of such skills in school. It is important to understand what works well and which incentives have no effects. The use of large-scale RCT’s in the field are of great importance to this. Yet these are not easy to develop, and the authors took the courage to undertake such a large-scale field experiment. The intervention, in turn, is an easy to implement one in educational practice if proven effective.

The study is well performed with a rigorous design and methodological approach, and the article is generally well-written. However, I think it can be sharpened a bit before publication. For example, I think the study can be somewhat bit stronger embedded in the literature. There is quite some research on the effectiveness of social-emotional learning programs or on the role of confidence nudging in relation to performance, that relates to this I guess. See some further comments below. I also recommend that the authors take a close look at how the information about the sample, procedures and measures is given, I sometimes had a difficult time grasping the details and keep the focus.

Comments per section

Introduction:

• p2, par3: The authors might want to take a look at a paper of Tenney et al. (2015) who investigate the relation between optimism and performance with a range of (small-scale) experiments, including some of which try to impact people’s optimism by encouragement/discouragement messages. For example, they observe that manipulated optimism affected their participants’ self-reports of felt optimism and a behavioral measure of their persistence, which are in turn important for performance.

Tenney et al., (2015) describe an experiment (Experiment 4) conducted via Amazon Mechanical Turk in which they manipulated young adults’ optimism by giving them random fictive performance feedbacks. This is a relevant study since it indicates that noncognitive skills are malleable and can be impacted by positive feedbacks. Like our results, Tenney et al find that experimentally induced noncognitive skills (optimism) do not lead to higher performance. We cite Tenney et al concerning these two arguments. However, we respectfully note that Tenney et al’ s paper was not conducted among students and, therefore, we provide a limited discussion of their results in our paper. 

• p2, par4: The authors mention that there is significant heterogeneity in the effect sizes observed in previous studies. I feel that they might elaborate a bit more these differences than is so far done in this paragraph. Now only the size and more general framework that the studies take are mentioned, but are there also some conclusions with respect to subgroups for example?

We have elaborated more on the paragraph you mentioned. The new paragraph reads as follows: prior meta-analyses show significant heterogeneity in the effect sizes; larger studies report a smaller effect size (Lösel & Beelmann, 2003). Programs introduced in education are particularly prone to a negative correlation between sample size and effect size (Slavin & Smith, 2009). Therefore, well-executed large-scale studies that employ an experimental design and impact students’ achievement via their noncognitive skills often report limited or no findings (Feron & Schils, 2020; Oreopoulos & Petronijevic, 2019). This suggests that small case studies are insufficient to determine a particular educational program’s scientific validity and practical utility. Therefore, upcoming large-scale studies should corroborate the explorative results of small-scale experiments and produce conclusive evidence of the effectiveness of a given program.

We have not provided more details about the specific results since the interventions in the related papers differ substantially from our light-touch intervention. Therefore, the subgroup-specific results of these papers are not conclusive for our results. 

• p2, par5: I am a bit puzzled by this argument. I understand that it might slow down the observed effects in studies, but in the end we want effective strategies to be integrated in teacher practice, right? I think you mean that the interventions proposed in the literature require more of an overhaul of the system. And that this might not always be feasible or desirable. But that there is a lack of studies on more easy to apply measures that could be integrated in education, independent of teacher motivation or experience. Perhaps I am misreading this, but the authors might want to explain a bit.

Thank you for raising this point. We streamlined our argument, and we have rewritten the paragraph. The efficacy of the developmental programs in education hinges on teachers’ understanding of the program and their capacity to implement it (Villase, 2014). These programs either require a change in teachers’ daily school routines or endow teachers with new skills. Altering teachers’ daily school routines can increase teachers’ workload. Teachers may thus become less motivated to implement these programs, ultimately inhibiting the program’s efficacy. Integrating developmental programs into teachers’ training systems and thus endowing teachers with new skills slow down the interventions’ return process (Duckworth et al., 2009). Only a scant number of studies propose light-touch interventions that are ready to be integrated into educational practice without requiring teachers’ motivation or experience.

• p2, par6: In my view there might be an additional concern prevalent when looking at the current literature and that is the lack of large-scale field RCT’s. Many of the experimental studies in these fields are either in a lab setting, or using small samples in the field if I am not mistaken. Studies in the field are mostly non- or quasi-experimental to my knowledge. The authors can correct me if I am wrong. Perhaps this might be added as an additional concern, also showing the contribution of this current paper.

We have included your argument in the sample size argument we raised, and we say that small case studies are insufficient to determine a particular educational program’s scientific validity and practical utility. Therefore, upcoming large-scale studies should corroborate the explorative results of small-scale experiments and produce conclusive evidence of the effectiveness of a given program.

• p3, par6: there is some discussion going on in the (education) literature on the effect sizes (small or null) in field experiments. The last two paragraphs of the discussion of a recent paper by Feron & Schils (2020) touch upon this issue and you might find this interesting for your study.

Thank you, we read the paper and discussed the implication in the discussion.

In short, encouraging students systematically and not just shortly before their exams is a possible school practice that can forge positive emotional involvement and engagement with the academic aspect of school life. Therefore, light-touch encouragement interventions might have substantial significance in themselves, even though these interventions do not directly affect students’ exam grades.

We interpret our results on the main treatment effects within the framework proposed by Jacob et al. (2019) of learning from null results. First, one should consider the typical potential growth in students’ exam grades over the intervention period. In our case, the intervention period is a couple of hours (i.e., the time elapsed between the time students received the message and the exam). Within such a short period, one should not expect large changes in students’ knowledge (that could be translated into higher grades). Therefore, the impact of any intervention (and not just particularly our encouragement campaign) that targets students a couple of hours before their exam might have a limited effect on students’ exam grades. Thus, the precisely estimated zero results in exam grades, which suggests that the intervention had no practical significance for students’ exam grades, could be attributed to the short period of time and (in addition) the light-touch (nonintensive) intervention. 

Second, one should consider the theory behind the outcomes. In our case, any change in students’ exam grades can be solely attributed to the change in a student’s ability belief targeted by the encouragement message. By contrast, changes in the secondary outcomes can be attributed to the encouraging words that students received in the treatment message. Therefore, our results indicate that the positive beliefs we experimentally induced by the encouragement intervention do not translate into higher cognitive performance in the short run. Nevertheless, encouraging words do affect self-efficacy. 

Lastly, one should consider the cost of the treatment. A low-cost intervention with a small impact might be considered successful despite the size of its impact, specifically due to the low costs. We invested about 210 USD in sending out the text messages; sending out the e-mails had no incidental costs. For this level of investment, a short-lived gain in students’ self-efficacy is a substantial achievement. Further, the implementation of the intervention does not require additional human effort; it could be scaled up to a virtually unlimited number of students. These features suggest that similar interventions can be worthwhile despite not directly boosting students’ exam grades. 

In sum, we conclude that automated encouragement messages shortly before students’ exams are not a panacea for increasing students’ academic achievement. However, students’ self-efficacy is sensitive to encouraging words, even if these words arrive shortly before an academically challenging exam situation. Thus, sending out encouraging messages shortly before students’ exams on a systematic rather than occasional basis might be a cost-effective tool for boosting students’ self-efficacy. Therefore, encouragement interventions might help to create a school climate that boosts students’ self-determination in the academic side of school life. They may thus have their own substantive importance (Appleton et al., 2008; Christenson et al., 2012). 

Design, data and method

• p4, par7: very minor query, but what kind of things can be bought in the SZTE gift shop? This might give some information about the incentive and to what extent it is a real incentive/reward.

Students could buy various products branded with the SZTE logo in the SZTE gift shop, like office supplies, mugs, t-shirts, sweatshirts, etc. The price of an average product is under 10,000 HUF. More information: https://szteshop.hu/en/

• p5, par6: Do you know how many students know that they did not receive the encouragement message? From those only 17% was sad/very sad, right? Is it the 33% mentioned in the next paragraph? This gives a bit more insight in the extent to which we can agree that that likelihood of adverse treatment effects is ‘moderate’, as you state.

We re-examined the treatment status after randomization at the end of the treatment period, when all messages had been sent out. We discovered that every student had received at least one e-mail message (before their first or second exam), but not every student had received the encouragement message (e.g., they only received the control message). 

Students did not receive the treatment message if their teachers entered the exam in question in the university’s registry after the exam had happened. In this case, we were not able to send students the encouragement message, since the corresponding exam was not listed in the university’s registry at that time. In sum, 3.65% of students (N = 565) did not receive an encouragement message. Our analysis is, therefore, an intention-to-treat (ITT) analysis.

• P7, par1: Perhaps you can already mention here that the first and second exams are in different subjects, because when I was reading this paragraph it was unclear to me why you did not distinguish between whether they received the message for the first or for the second exam? The information about the differences between the first and second exams, as well as information on the general exam system in Hungary follows later, but the reader might already be a bit puzzled. It is many details to digest.

Thank you very much for this comment. We included this argument: The first and second exams are in different subjects—this difference is controlled for in the analysis.

• p8, par 1: how much time is there between the pre- and post-test? Is it a reasonable period to expect effects?

Motivated by your suggestion, we described our design as more focused. 

Figure 1 shows when students completed the endline survey relative to the corresponding exam. On average, students filled in the questionnaire 13 hours before their exam. This means that the treatment e-mail targeted the students a couple of hours before their exam. 

Figure 2 shows the time (in hours) relative to the exam when the treatment SMS was sent out to students’ mobile devices. The majority of students (66%) received the treatment SMS 3 hours before the exam, indicating that we encouraged students shortly before their exams. 

In the discussion, we acknowledge that the intervention period is a couple of hours (i.e., the time elapsed between the time students received the message and the exam). Within such a short period, one should not expect large changes in students’ knowledge (that could be translated into higher grades). Therefore, the impact of any intervention (and not just particularly our encouragement campaign) that targets students a couple of hours before their exam might have a limited effect on students’ exam grades. Thus, the precisely estimated zero results in exam grades, which suggests that the intervention had no practical significance for students’ exam grades, could be attributed to the short period of time and (in addition) the light-touch (nonintensive) intervention. 

• p8, par 3, you might not know, but might the missings due to illness be related to test anxiety? If you have any information on this, that would be useful, e.g. perhaps those that scored high on test anxiety in the survey are more often absent?

Highly anxious students with low self-confidence might be more likely to report illness, which could cause selective attrition in the primary outcome. We tested these hypotheses in a study-program fixed effect bivariate linear probability model. We found that neither baseline text anxiety (p = 0.7) nor baseline self-confidence (p = 0.28) is associated with missingness in the primary outcome. Thank you very much for this suggestion. 

• p8, footnote: ether > either.

Thank you, we have corrected the typo.

• p9, point 3: I am bit surprised by the locus of control, this is not mentioned in the literature. Perhaps the authors can address it in the literature, so the reader understands why it is included.

Thank you for this highly constructive suggestion. We have clarified why we measure locus of control as a baseline variable. We write that locus of control measures the sense of agency people feel over their lives. Locus of control is believed to be conceptually similar to self-efficacy (Rotter, 1992) and is conceptually connected to behavioral intention and control in Ajzen’s theory of planned behavior (Ajzen, 2002). We measured the baseline external/internal locus of control (Rotter 1966) using the four-item version of the Rotter-scale test (Andrisani, 1977; Goldsmith et al., 1996). In the test, respondents choose between two sentences describing external and internal control conditions. People with an internal locus of control believe that their abilities and actions influence their life outcomes. By contrast, people with an external locus of control believe that random chance and environmental factors affect their life outcomes.

Results

• p13, par 6: hypostatized > hypothesized.

Thank you, we have corrected the typo.

Discussion

• p15, par 5 and p16, par 6: I was just wondering about the effect of the treatment on exam grades, these are only given in 1 2 3 4 5, right? In that case the treatment should be really strong to see an effect on grades? Or am I misinterpreting the grading system? It might be that in the previous literature the grading system used was different and allowed for ‘easier to establish’ effects?

The primary outcome variable is students’ exam grades, measured in integers between 1 and 5. Grade 1 means fail. Other grades are equivalent to passing the exam, and in ascending order they express the quality of students’ performance, with 5 as the best. Relative grading is used in Hungary; that is, there is no absolute benchmark to which teachers relate students’ performance.

We acknowledge in the section describing the outcome variables that our primary outcome can take only five values. Thus the chances to find significant treatment effects on students’ exam grades are smaller than finding significant treatment effects on the secondary outcomes since these variables range between 0 and 10.

• p15, par 6 and later when you discuss this more thorougly in the discussion: this result for the high able students might indeed link up to boosting confidence that increases the grades. (it relates to the general effect on self-confidence, you observe). They already knew they were good (or among the upper part of the ability distribution) and receiving an encouragement message basically confirms that feeling and they even get more confident in that they will succeed in the exam. Perhaps the psychological literature on (over)confidence might be useful here, you might want to check out papers of Don Moore, who wrote about this. The low ability students might indeed have given up, and have become rather ignorant to studying and performing well on tests. While it is quite important also for their future training participation as many studies show that low-educated/ability workers are less prone to investing in further training during the life course. More emphasis might be put on understanding the mechanisms behind the non-effects of encouragement among low-ability students. However, having said that, I think the conclusions on the heterogeneity by ability should be modest, as the observed effects were only marginally significant. Moreover, we are talking about low-ability students in a university setting, so not overall low-ability students, i.e. those that already made it to an academic study. I think this is important to mention.

We have incorporated this argument in the discussion when we are discussing the heterogeneous treatment effect in exam grades. We write that students with lower baseline abilities may have less confidence in their abilities (Wigfield, 1994). Therefore, they might not believe that the encouragement message is addressed to them. In particular, students with lower ability may achieve lower grades at university. They could falsely conclude that they are not successful and regard the message as not relevant. By contrast, more able students who achieve better grades might subjectively rate themselves as more successful and therefore place greater trust in the encouragement message. 

• p16, par 6: perhaps the effect on the noncognitive skills needs more time to translate to cognitive skills, have you considered that? I would at least say it did not translate into short-run cognitive results.

We cannot say this explicitly since we found significant main effects on self-efficacy. However, we have sharpened our argument in the discussion, and we write that our results suggest that self-efficacy is malleable and can be impacted by the positive feedback received independently of one’s performance (Bouffard-Bouchard 1990; Tenney et al. 2015). However, the development of students’ test anxiety or motivation requires a different treatment. 

I hope the authors can use my comments and suggestions to further improve the paper.

Thank you very much for your careful reading and valuable comments!

---

## [Decision Letter · Decision Letter 1]

30 Jun 2021

PONE-D-21-03213R1

Not just words! Effects of a light-touch randomized encouragement intervention on students’ exam grades, self-efficacy, motivation, and test anxiety

PLOS ONE

Dear Dr. Keller,

Thank you for submitting your manuscript to PLOS ONE. After careful consideration, we feel that it has merit but does not fully meet PLOS ONE’s publication criteria as it currently stands. Therefore, we invite you to submit a revised version of the manuscript that addresses the points raised during the review process.

The paper has significantly improved in the current version review. However, there are still some important points  that need to be fixed. As two referees point out, the effect of the parameter beta3 is ambiguous. If this problem is fixed, it may change the cost-benefit discussion, as suggested by Reviewer #1. Thus, the authors should carefully clarify the points raised by Reviewers #1 and #2.

We look forward to receiving your revised manuscript.

Kind regards,

Alfonso Rosa Garcia

Academic Editor

PLOS ONE

Reviewers' comments:

Reviewer's Responses to Questions

**Comments to the Author**

1. If the authors have adequately addressed your comments raised in a previous round of review and you feel that this manuscript is now acceptable for publication, you may indicate that here to bypass the “Comments to the Author” section, enter your conflict of interest statement in the “Confidential to Editor” section, and submit your "Accept" recommendation.

Reviewer #1: (No Response)

Reviewer #2: (No Response)

Reviewer #3: All comments have been addressed

2. Is the manuscript technically sound, and do the data support the conclusions?

Reviewer #1: Partly

Reviewer #2: Yes

Reviewer #3: Yes

3. Has the statistical analysis been performed appropriately and rigorously? 

Reviewer #1: Yes

Reviewer #2: Yes

Reviewer #3: Yes

4. Have the authors made all data underlying the findings in their manuscript fully available?

Reviewer #1: Yes

Reviewer #2: Yes

Reviewer #3: Yes

5. Is the manuscript presented in an intelligible fashion and written in standard English?

Reviewer #1: Yes

Reviewer #2: Yes

Reviewer #3: Yes

6. Review Comments to the Author

Reviewer #1: This version of the paper is much improved. I enjoyed re-reading it.

I thank the authors for addressing my comments. It is now clearer how they selected the message content, when students received both the email and text messages, and that the length of text messages was fixed. It also now easier for the reader to interpret the parameter, beta 3, as the mean difference in exam grades between treated and control students in the second exam minus this same difference in the first exam. The authors have also done a nice job of toning down the text throughout the paper to better reflect the findings and have nicely explained how they are thinking about the cost-benefit analysis of their intervention.

While these are all great improvements, I do have three remaining concerns about the paper.

*First*, while the mathematical representation of the parameter, beta 3, is now clear, I still do not quite understand how to think about it in the context of a carryover effect. This parameter is negative in Tables 2 to 4, which means that the difference in outcomes between treated and control students is higher on the first exam than it is on the second exam. Why do the authors think this is the case? What is the hypothesized mechanism?

In Table 4, the estimate of beta 3 is negative and, together with beta 1, implies that treatment was ineffective in group B. Is this because the treatment on group A had a persistent effect on exam 2, pushing up the exam 2 scores of group A (relative to what would have been the case without treatment on exam 1)? Or is it because treatment is simply more effective when applied earlier in the semester? It is still unclear to me how to think about the underlying dynamics that result in the estimated value for this parameter. I think the authors could provide more of an explanation than is currently provided between lines 599 and 605 of the manuscript.

*Second*, and related, these issues around beta 3 become especially relevant when it comes to assuaging concerns about selection into the endline questionnaire. The authors say treatment status decreases completion of the endline survey but that Tables A6 to A8 show similar results when the sample is restricted to students who completed the endline survey twice. I respectfully disagree that the results are similar.

In particular, Table 3 is one of the most important tables in the paper, highlighting the only effect on one of the secondary outcomes—namely, self-efficacy. (The authors concede that the effect on motivation is less clear because it does not replicate on exam 2.) In Table 3, the treatment effect is large and present on both exam 1 and exam 2. But in Table A7, the analogue to Table 3 but with only students who completed the endline survey twice, treatment appears to influence self-efficacy only on exam 1. That is, beta 3 is negative and significant in nearly all the first seven columns, and the sum of beta 1 and beta 3 is considerably smaller than just beta 1. I cannot tell when the sum remains statistically significant, but it surely does not in all specifications and the resulting treatment effect on exam 2 is always much smaller than the estimate of beta 1 in Table 3. I read the contrast between Tables 3 and A7 as potentially indicating that selection into the endline survey by treatment status is a problem, as Table A7 provides another instance (in addition to Table 4) of the treatment effect not replicating on exam 2.

*Third*, while the authors have done a great job explaining their cost-benefit analysis, I am not convinced that this is a program worth scaling—or at least that this paper provides evidence to that effect.

To start, there was no effect on exam grades. This may because there simply was not enough time between the encouragement and the exam for students to change behavior, as the authors point out in the discussion. The authors also note on lines 705 and 706 that grading on a curve may be the reason why exam grades did not go up. It seems that neither explanation warrants an expansion of the program the authors tested. If enough time had not passed between encouragement and the exam, then one should consider an intervention that encourages students earlier or more consistently throughout the semester. But that is not the intervention about which this paper presents evidence. If grading on a curve prevents an effect on exams, then it is unclear how any intervention might work.

I agree with the authors that exam grades are not the only (or even the most important) outcome worth considering. The secondary outcomes the authors test are also important. But, as mentioned, I am not convinced that treatment did influence any of the secondary outcomes that the authors explore. Another, related encouragement campaign might, but again, that evidence is not presented here. In sum, I see very little evidence for the benefit side of the cost-benefit analysis for the program studied in this paper.

Reviewer #2: 1. Thank you for your clarifications. I am still confused about how you are describing your models. As I understand it, you had two groups, Group A and Group B. Group A was treated on test 1, Group B was treated on test 2 (except in the small percentage of cases where this did not occur).

Your basic model is specified as:

Y=beta0+beta1XTreatment+beta2XTest 2+beta3XTreatmentXTest 2+epsilon.

As far as I understand it the way you have specified the Treatment variable is 1 for group A for test 1, 0 for group A for test 2, 0 for group B for test 1, 1 for group B for test 2. Is this correct?

The way you describe a carryover effect it is "a significant carry-over effect reflects that encouraging students before their first exam affects their grades at the second exam." I think this is one plausible interpretation of beta3 but I do not think it is the only plausible explanation for possible differences. The only accurate way to describe the effects I think is as the difference in the effects between the first and second exam. It could be that the effect differences come from the ordering of the treatments as you say, but another possibility is that there is a difference in the effect due to the difficultly or nature of the first versus the second exam. Perhaps these messages have a different effect later in the semester. Perhaps the messages have more effect on one exam because of the content of one exam versus the other.

Reviewer #3: The authors have responded pretty well to the comments I raised. I am satisfied with the revisions made.

I think the manuscript improved due to this revision. I actually do not have any further comments.

7. PLOS authors have the option to publish the peer review history of their article (what does this mean?). If published, this will include your full peer review and any attached files.

Reviewer #1: No

Reviewer #2: **Yes: **Daniel Dench

Reviewer #3: **Yes: **prof. dr. trudie schils

---

## [Author Response · Author response to Decision Letter 1]

26 Jul 2021

Reviewer #1: 

This version of the paper is much improved. I enjoyed re-reading it.

I thank the authors for addressing my comments. It is now clearer how they selected the message content, when students received both the email and text messages, and that the length of text messages was fixed. It also now easier for the reader to interpret the parameter, beta 3, as the mean difference in exam grades between treated and control students in the second exam minus this same difference in the first exam. The authors have also done a nice job of toning down the text throughout the paper to better reflect the findings and have nicely explained how they are thinking about the cost-benefit analysis of their intervention.

While these are all great improvements, I do have three remaining concerns about the paper.

*First*, while the mathematical representation of the parameter, beta 3, is now clear, I still do not quite understand how to think about it in the context of a carry-over effect. This parameter is negative in Tables 2 to 4, which means that the difference in outcomes between treated and control students is higher on the first exam than it is on the second exam. Why do the authors think this is the case? What is the hypothesized mechanism?

In Table 4, the estimate of beta 3 is negative and, together with beta 1, implies that treatment was ineffective in group B. Is this because the treatment on group A had a persistent effect on exam 2, pushing up the exam 2 scores of group A (relative to what would have been the case without treatment on exam 1)? Or is it because treatment is simply more effective when applied earlier in the semester? It is still unclear to me how to think about the underlying dynamics that result in the estimated value for this parameter. I think the authors could provide more of an explanation than is currently provided between lines 599 and 605 of the manuscript.

Thank you very much for raising this important point which led us to expand the explanation on the carry-over effect. 

The interaction of T and E indicates the carry-over effect, i.e., whether the ordering of the treatment influences the outcome variables. A significant carry-over effect biases the estimation of the average treatment effect. 

In our design, we expect a negative carry-over effect, which means that encouraging students before their first exam affects their outcomes at the second exam. Since the sequence of treated and control conditions is either treated-control (Group A) or control-treated (Group B), treating students first might lead to a long-lasting effect or a long wash-out period. A statistically significant negative carry-over effect signals that the treatment effect is higher at students’ first exam than at their second. A negative carry-over effect legitimizes the encouragement treatment and shows that students yearn for encouragement, since treating them before their first exam also affected their outcomes at the second exam, when they were not treated. 

Under the current design, the carry-over effect does not provide a substantive interpretation of possible mechanisms that might lead to the longer-lasting effect when treating students before the first exam instead of the second.

*Second*, and related, these issues around beta 3 become especially relevant when it comes to assuaging concerns about selection into the endline questionnaire. The authors say treatment status decreases completion of the endline survey but that Tables A6 to A8 show similar results when the sample is restricted to students who completed the endline survey twice. I respectfully disagree that the results are similar.

In particular, Table 3 is one of the most important tables in the paper, highlighting the only effect on one of the secondary outcomes—namely, self-efficacy. (The authors concede that the effect on motivation is less clear because it does not replicate on exam 2.) In Table 3, the treatment effect is large and present on both exam 1 and exam 2. But in Table A7, the analogue to Table 3 but with only students who completed the endline survey twice, treatment appears to influence self-efficacy only on exam 1. That is, beta 3 is negative and significant in nearly all the first seven columns, and the sum of beta 1 and beta 3 is considerably smaller than just beta 1. I cannot tell when the sum remains statistically significant, but it surely does not in all specifications and the resulting treatment effect on exam 2 is always much smaller than the estimate of beta 1 in Table 3. I read the contrast between Tables 3 and A7 as potentially indicating that selection into the endline survey by treatment status is a problem, as Table A7 provides another instance (in addition to Table 4) of the treatment effect not replicating on exam 2.

We would like to express to you our highest gratitude for the careful reading of the Appendix tables. We realized that we made a severe failure when we copied the results from the output files to the paper. Specifically, we exchanged the exam effect with the carry-over effect. Therefore, your reading of the results based on our wrongly edited paper was correct! 

We have cleaned this inconsistency, and the correct appendix tables now show no carry-over effect. Specifically, the results shown in Table 3 and Table A7 are qualitatively similar. The tables can be reproduced and checked since we provided the data and analytical scripts at the OSF platform cited in the paper. 

Your comment has inspired us to edit all tables. We included the treatment’s effect concerning the second treatment since it is the linear combination of the coefficients β_1 and β_3. The last row of tables (Table 3-5 and Tables A4-A8) now contains the treatment effect concerning students’ second exam, with the corresponding standard errors. 

*Third*, while the authors have done a great job explaining their cost-benefit analysis, I am not convinced that this is a program worth scaling—or at least that this paper provides evidence to that effect.

To start, there was no effect on exam grades. This may because there simply was not enough time between the encouragement and the exam for students to change behavior, as the authors point out in the discussion. The authors also note on lines 705 and 706 that grading on a curve may be the reason why exam grades did not go up. It seems that neither explanation warrants an expansion of the program the authors tested. If enough time had not passed between encouragement and the exam, then one should consider an intervention that encourages students earlier or more consistently throughout the semester. But that is not the intervention about which this paper presents evidence. If grading on a curve prevents an effect on exams, then it is unclear how any intervention might work.

I agree with the authors that exam grades are not the only (or even the most important) outcome worth considering. The secondary outcomes the authors test are also important. But, as mentioned, I am not convinced that treatment did influence any of the secondary outcomes that the authors explore. Another, related encouragement campaign might, but again, that evidence is not presented here. In sum, I see very little evidence for the benefit side of the cost-benefit analysis for the program studied in this paper.

Thank you very much for this very insightful argument, which motivated us to delate the policy recommendation concerning scaling up the treatment. 

We have reframed our conclusion and write that light-touch, automated encouragement messages, requiring minimal additional human effort from the message provider and sent shortly before exams, do not affect students’ exam grades. Nevertheless, we have isolated a possible mechanism through which encouragement interventions might exert their effect. Specifically, we found that self-efficacy is sensitive to encouraging words, even if students only receive them on an occasional basis shortly before an academically challenging exam situation. Therefore, further encouragement interventions targeting students’ self-efficacy might promote a school climate that boosts students’ engagement in the academic side of school life.

Nevertheless, we made it clear that future encouragement interventions should further improve on our automated encouragement message, which required minimal additional human effort from the message provider. For example, personalized (rather than uniform) messages sent by senders to whom students have contact (e.g., a professor or role model rather than the Head of the Directorate of Education, with whom most students do not have direct contact) could increase the efficacy of future treatments. Furthermore, interventions that encourage students earlier, or more consistently throughout the semester on a systematic rather than occasional basis, should also be considered to increase the treatment effect. 

 

Reviewer #2: 

1. Thank you for your clarifications. I am still confused about how you are describing your models. As I understand it, you had two groups, Group A and Group B. Group A was treated on test 1, Group B was treated on test 2 (except in the small percentage of cases where this did not occur).

Your basic model is specified as:

Y=beta0+beta1XTreatment+beta2XTest 2+beta3XTreatmentXTest 2+epsilon.

As far as I understand it the way you have specified the Treatment variable is 1 for group A for test 1, 0 for group A for test 2, 0 for group B for test 1, 1 for group B for test 2. Is this correct?

Thank you very much for your consideration, your interpretation is correct. 

The way you describe a carry-over effect it is “a significant carry-over effect reflects that encouraging students before their first exam affects their grades at the second exam.” I think this is one plausible interpretation of beta3 but I do not think it is the only plausible explanation for possible differences. The only accurate way to describe the effects I think is as the difference in the effects between the first and second exam. It could be that the effect differences come from the ordering of the treatments as you say, but another possibility is that there is a difference in the effect due to the difficultly or nature of the first versus the second exam. Perhaps these messages have a different effect later in the semester. Perhaps the messages have more effect on one exam because of the content of one exam versus the other.

Thank you very much for raising this important point which led us to expand the explanation on the carry-over effect.

The interaction of T and E indicates the carry-over effect, i.e., whether the ordering of the treatment influences the outcome variables. A significant carry-over effect biases the estimation of the average treatment effect. 

In our design, we expect a negative carry-over effect, which means that encouraging students before their first exam affects their outcomes at the second exam. Since the sequence of treated and control conditions is either treated-control (Group A) or control-treated (Group B), treating students first might lead to a long-lasting effect or a long wash-out period. A statistically significant negative carry-over effect signals that the treatment effect is higher at students’ first exam than at their second. A negative carry-over effect legitimizes the encouragement treatment and shows that students yearn for encouragement, since treating them before their first exam also affected their outcomes at the second exam, when they were not treated. 

Under the current design, the carry-over effect does not provide a substantive interpretation of possible mechanisms that might lead to the longer-lasting effect when treating students before the first exam instead of the second.

We respectfully note that the exam dummy captures all differences concerning students’ first and second exams, including the difference in the first versus second exams’ difficultly or nature. 

We clarified that students took their second exam soon after their first exam. The median student had four days between their first and second exams, and most frequently (in 21% of cases), there was only one day between the two exams.

 

Reviewer #3:

The authors have responded pretty well to the comments I raised. I am satisfied with the revisions made.

I think the manuscript improved due to this revision. I actually do not have any further comments.

Thank you very much for this positive assessment and evaluation.

---

## [Decision Letter · Decision Letter 2]

20 Aug 2021

Not just words! Effects of a light-touch randomized encouragement intervention on students’ exam grades, self-efficacy, motivation, and test anxiety

PONE-D-21-03213R2

Dear Dr. Keller,

We’re pleased to inform you that your manuscript has been judged scientifically suitable for publication and will be formally accepted for publication once it meets all outstanding technical requirements.

Kind regards,

Alfonso Rosa Garcia

Academic Editor

PLOS ONE

Additional Editor Comments (optional):

Reviewers' comments:

Reviewer's Responses to Questions

**Comments to the Author**

1. If the authors have adequately addressed your comments raised in a previous round of review and you feel that this manuscript is now acceptable for publication, you may indicate that here to bypass the “Comments to the Author” section, enter your conflict of interest statement in the “Confidential to Editor” section, and submit your "Accept" recommendation.

Reviewer #1: All comments have been addressed

Reviewer #2: All comments have been addressed

2. Is the manuscript technically sound, and do the data support the conclusions?

Reviewer #1: Yes

Reviewer #2: Yes

3. Has the statistical analysis been performed appropriately and rigorously? 

Reviewer #1: Yes

Reviewer #2: Yes

4. Have the authors made all data underlying the findings in their manuscript fully available?

Reviewer #1: Yes

Reviewer #2: Yes

5. Is the manuscript presented in an intelligible fashion and written in standard English?

Reviewer #1: Yes

Reviewer #2: Yes

6. Review Comments to the Author

Reviewer #1: Thank you for your detailed edits and replies. I am glad to see that results in the appendix tables were previously incorrect and that the authors were able to correct these mistakes. These new, correct results make the paper stronger, internally consistent, and help assuage concerns about selection into the endline questionnaire (as the authors originally intended). The edits to the conclusion are also welcome, as I believe they better reflect the paper’s findings.

However, I still believe the authors are limited in what they can say about carry-over effects. While the new draft improves upon the discussion of carry-over effects, I still have some of the same concerns:

1.The authors mentioned a significant carry-over effect biases the estimation of the average treatment effect. This is only the case when trying to estimate the treatment effect on the second exam because there is no pure control group in this case. A significant carry-over effect from exam 1 to 2, if it exists, is part of the overall treatment effect (not a source of bias if a pure control group were to exist).

2.More importantly, the pure carry-over effect is simply not identified in the authors setting. Identification of a carry-over or persistent effect would require a pure control group (untreated on both exam 1 and exam 2) whose exam 2 score could be used as the baseline with which to compare the exam 2 score of group A.

In any event, the estimates of beta_3 do not seem to play a big role in the authors main results or message any longer (with the exception of Table 4), so I am not as concerned about these limitations as with previous drafts.

Again, I commend the authors on all the improvements they have made and enjoyed reading the paper.

Reviewer #2: Although I still disagree with the interpretation of the carry-over effect, I don't want to hold up publication on this account alone as your interpretation is one plausible interpretation of the effect you find.

You say:

"We respectfully note that the exam dummy captures all differences concerning

students’ first and second exams, including the difference in the first versus second

exams’ difficultly or nature."

But the interaction effect is the difference in the effect of your intervention on the outcomes from the first to the second test. My suggestion was that the effect of your intervention could also be different because of the difference in difficulty or nature of the second test. This would be unrelated to whether the effect carries over and this effect would be included in the interaction term, not the exam dummy. If you think these things are not plausible given your much closer read of the data then I accede to your interpretation.

7. PLOS authors have the option to publish the peer review history of their article (what does this mean?). If published, this will include your full peer review and any attached files.

Reviewer #1: No

Reviewer #2: **Yes: **Daniel Dench

---

## [Editor Report · Acceptance letter]

6 Sep 2021

PONE-D-21-03213R2 

Not just words! Effects of a light-touch randomized encouragement intervention on students’ exam grades, self-efficacy, motivation, and test anxiety 

Dear Dr. Keller:

I'm pleased to inform you that your manuscript has been deemed suitable for publication in PLOS ONE. Congratulations! Your manuscript is now with our production department. 

Kind regards, 

on behalf of

Dr. Alfonso Rosa Garcia 

Academic Editor

PLOS ONE